# Precise excision of expanded GGC repeats in *NOTCH2NLC* via CRISPR/Cas9 for treating neuronal intranuclear inclusion disease

Nina Xie[1,2,9], Yongcheng Pan [2,9], Huichun Tong[3], Yingqi Lin[3], Ying Jiang[4], Zhiqin Wang[1,2], Juan Wan[5], Wendiao Zhang [5], Xinhui Wang[2], Xiaobo Sun[6], Sen Yan [3], Peng Yin [3], Qiying Sun[1], Chengzhi Qi[6], Yun Tian[1], Lu Shen [2,7], Hong Jiang[2,7], Desheng Liang [4], Beisha Tang [2,5], Shihua Li [3] ✉, Xiao-Jiang Li [3,8] ✉ & Qiong Liu [2] ✉

Neuronal intranuclear inclusion disease (NIID) is an adult-onset neurodegenerative disease caused by expanded GGC repeats in the 5′ untranslated region of the human-specific *NOTCH2NLC* gene. The high sequence similarity between *NOTCH2NLC* and its paralogs poses a significant challenge for precise gene editing. Here, we develop a CRISPR/spCas9-based gene-editing strategy that precisely excises the expanded GGC repeats in *NOTCH2NLC* without detectable off-target effects on the highly homologous *NOTCH2/NOTCH2NL* family genes (<2% sequence divergence at this locus). The efficacy, specificity and safety of this approach are rigorously validated across multiple experimental models, including human cell lines, NIID iPSCs, and our previously established transgenic NIID mouse model. Our results demonstrate that precise excision of the expanded GGC repeats effectively alleviates NIID-related neuropathological, molecular and behavioral abnormalities. This study establishes the proof of concept for genome editing as a therapeutic strategy for NIID and other related repeat expansion disorders.

Repeat expansion disorders (REDs) represent a class of genetic diseases characterized by the pathological expansion of tandem repeat sequences beyond their normal range[1,2]. To date, more than sixty REDs have been identified in humans, with the majority manifesting as neurodevelopmental or neurodegenerative diseases, such as fragile X syndrome, Huntington's disease, spinocerebellar ataxia and *C9ORF72*-related amyotrophic lateral sclerosis and frontotemporal dementia[1,3–8]. Notably, recent advancements in long-read sequencing and whole-genome sequencing (WGS) technologies have significantly accelerated the discovery of new REDs[9–12]. Among these, GGC repeat expansions located within the "non-coding" regions of the human genome have garnered particular attention[13]. One such disorder, neuronal intranuclear inclusion disease (NIID) has attracted increasing attention in recent years.

NIID is a progressive neurodegenerative disorder characterized by widespread eosinophilic intranuclear inclusions in the nervous system and visceral organs[14,15]. It exhibits substantial clinical heterogeneity, with

[1]Department of Geriatrics, Xiangya Hospital, Central South University, Changsha, Hunan, China. [2]Key Laboratory of Hunan Province in Neurodegenerative Disorders & Department of Neurology, Xiangya Hospital, Central South University, Changsha, Hunan, China. [3]Guangdong Provincial Key Laboratory of Non-human Primate Research, Guangdong-Hongkong-Macau Institute of CNS Regeneration, Jinan University, Guangzhou, China. [4]Centre for Medical Genetics & Hunan Key Laboratory of Medical Genetics, School of Life Sciences, Central South University, Changsha, Hunan, China. [5]Department of Neurology, Multi-Omics Research Center for Brain Disorders, The First Affiliated Hospital, Hengyang Medical School, University of South China, Hengyang, Hunan, China. [6]School of Statistics and Mathematics, Zhongnan University of Economics And Law, Wuhan, Hubei, China. [7]National Clinical Research Center for Geriatric Disorders, Xiangya Hospital, Central South University, Changsha, Hunan, China. [8]Lingang Laboratory, Shanghai, China. [9]These authors contributed equally: Nina Xie, Yongcheng Pan. ✉e-mail: lishihualis@jnu.edu.cn; xjli33@jnu.edu.cn; lqiong66@csu.edu.cn

primarily neurological manifestations including dementia, muscle weakness, tremor, bradykinesia, cerebellar ataxia, periodic encephalitic episodes, and autonomic dysfunction[9,14–18], as well as non-neurological involvement[19–21]. Severe cases can lead to loss of ambulation and premature death[14,17,18]. In 2019, the GGC repeat expansion (>60 repeats) in the 5′ untranslated region (5′UTR) of the *NOTCH2NLC* gene was reported as the genetic cause of NIID by several groups[9,22–24]. Furthermore, this mutation has also been implicated in a series of neurodegenerative and neuromuscular diseases[22,25–32]. Although comprehensive epidemiological data are limited, over 600 cases have been documented worldwide by the end of 2022[33]. Despite these genetic insights, no effective treatments currently exist, and conventional pharmacological therapies provide only limited symptomatic relief without halting disease progression.

Previous studies have confirmed that the GGC repeat expansions in the *NOTCH2NLC* gene are located within an independent upstream open reading frame (uORF) positioned before the canonical ATG, and these expanded GGC repeats can be translated into multiple toxic polypeptides, with polyglycine (polyG) being the predominant one[34]. These toxic peptides accumulate as intranuclear inclusions in the central nervous system and visceral organs, driving disease pathology through mechanisms such as dysregulated alternative splicing, impaired nuclear-cytoplasmic transport, mitochondrial dysfunction, and abnormal ribosome biogenesis[35–41]. Consequently, targeting the expanded GGC repeats through gene editing represents a rational and potential therapeutic strategy for NIID.

The human-specific *NOTCH2NL* (*NOTCH2* N-terminal like) gene family, which includes *NOTCH2NLA*, *NOTCH2NLB*, *NOTCH2NLC* and *NOTCH2NLR*, shares high sequence homology and plays critical roles in human brain development and homeostasis[42–44]. Specifically, *NOTCH2NLA*, *NOTCH2NLB*, and *NOTCH2NLC* are expressed in cortical progenitors, where they enhance self-renewal and delay premature differentiation by potentiating canonical Notch signaling, thereby contributing to the evolutionary expansion of the human cerebral cortex. Abnormal dosage of *NOTCH2NL* genes has been linked to neurodevelopmental disorders, including microcephaly or cortical malformations upon loss and macrocephaly or autism spectrum disorders upon duplication. Therefore, it is essential to specifically target the GGC repeat expansion within *NOTCH2NLC* while preserving the normal expression and function of the *NOTCH2NL* family. However, the near-identical sequence flanking the GGC repeats (<2% divergence) among these homologous genes, combined with the large size and high GC content of the expanded repeats, pose significant challenges for precise and efficient gene editing.

In this study, we develop a CRISPR/spCas9-based gene-editing strategy to specifically target the GGC repeat expansions within the *NOTCH2NLC* gene. To ensure editing precision, we design sgRNAs with minimal sequence homology to *NOTCH2NL* paralogs and *NOTCH2* and employed a dual-sgRNA approach. We systematically assess its efficacy, specificity and safety across multiple models, including HEK293 cells with normal GGC repeats or expanded GGC repeats in *NOTCH2NLC*, NIID patient-derived induced pluripotent stem cells (iPSCs) and neural progenitor cells (NPCs), as well as a previously established transgenic NIID mouse model. Importantly, targeted excision of the expanded GGC repeats markedly reduces polyG protein levels and significantly ameliorates neuropathological, molecular and behavioral phenotypes associated with NIID. Collectively, our work establishes a strong preclinical foundation for the development of genome-editing-based therapies for NIID.

## Results

### CRISPR/Cas9-based gene-editing specifically targeting GGC repeats within *NOTCH2NLC* in human cell lines with normal GGC repeats

Given the near-identical sequence flanking the GGC repeats among *NOTCH2NLC*, *NOTCH2* and other members of the *NOTCH2NL* family

(*NOTCH2NLA*, *NOTCH2NLB*, and *NOTCH2NLR*, *NOTCH2NLA/B/R* for short), we conducted comparative analysis and identified two mismatch regions flanking the GGC repeats (Supplementary Fig. 1A). To enable precise excision of the expanded GGC repeats of *NOTCH2NLC*, we designed multiple single guide RNAs (sgRNAs) targeting the upstream and downstream mismatch regions. Specifically, sgRNA 1-4 were designed to target sequences upstream of the GGC repeats, while sgRNA 5-8 targeted sequences downstream (Fig. 1A, Supplementary Fig. 1B). The details of sgRNAs are provide in Supplementary Data 1.

First, the gene-editing efficiency of each sgRNA was evaluated in human cell lines with normal GGC repeats (HEK293 cells). Forty-eight hours after co-transfection with Cas9 and sgRNA plasmids (Supplementary Fig. 1C) at a Cas9-to-single sgRNA ratios of 4:1, which was selected for preliminary screen based on our prior experience in genome editing of other genes[45]. We performed a T7 endonuclease I (T7EI) assay on the PCR products amplified using *NOTCH2NLC*-specific primers (Supplementary Fig. 1A and Supplementary Data 2). The results indicated that the upstream sgRNA1, as well as the downstream sgRNA5 and sgRNA6, exhibited optimal cutting efficiency (Fig. 1B–E). Intriguingly, unexpected T7EI cleavage fragments (400–500 bp) were observed in the control samples without CRISPR/Cas9 targeting (Fig. 1B, C). Since the extracted gDNA pooled multiple HEK293 cells, the unexpected bands may reflect intrinsic variations in the GGC repeat length of *NOTCH2NL* among cells within the same population (Supplementary Fig. 2A).

To validate the efficacy of these sgRNAs mediating GGC repeats deletion, various sgRNA combinations were tested. T7EI assay performed on the PCR products amplified using *NOTCH2NLC*-specific primers demonstrated that both sgRNA1 + 5 and sgRNA1 + 6 combinations efficiently excised the GGC repeats (33–40%), as evidenced by truncated PCR products due to the GGC deletion (Fig. 1F–G) and mismatch cleavage products (Supplementary Fig. 2B). For each combination, we tested different Cas9-to-sgRNA combinations ratios (1:1, 2:1, and 4:1) to optimize the targeting efficiency. The 1:1 ratio was found to be the most effective (Supplementary Fig. 2C) and was subsequently used in all following experiments.

Then, we evaluated whether the Cas9-mediated deletion of GGC repeats was specific to *NOTCH2NLC* in HEK293 cells. Genomic DNA region encompassing GGC repeats was amplified using consensus primers that amplify *NOTCH2NLA/B/C/R* and *NOTCH2* simultaneously (Supplementary Fig. 1A). The PCR products were subjected to TA cloning, and fifty randomly selected colonies per sgRNA combination were subjected to Sanger sequencing. The results confirmed that the deletion of GGC repeats occurred exclusively in the *NOTCH2NLC* gene, with no detectable off-target editing in homologous *NOTCH2NLA/B/R* and *NOTCH2* (Fig. 1H, I). The specificity of gene-editing was further validated by PCR using primers that specifically amplify GGC repeats region of *NOTCH2NLA/B/R* and *NOTCH2*. The agarose gel electrophoresis confirmed the absence of truncated PCR products or deletions in these homologous genes (Fig. 1J and Supplementary Fig. 2D). In addition, quantitative PCR (qPCR) analysis using *NOTCH2NLC*-specific primers, or consensus primers, which amplify *NOTCH2NLA/B/C/R* and *NOTCH2*, demonstrated that the deletion of GGC repeats did not alter the expression of *NOTCH2NLC* or its homologs (Supplementary Fig. 3A), providing additional evidence for the specificity of the gene-editing strategy.

### Gene-editing of expanded GGC repeats significantly reduced polyG levels in stable HEK293 cell line expressing *NOTCH2NLC*-98GGC

Accumulating evidence supports that the polyG polypeptides, translated from the expanded GGC repeats, play a central role in the pathogenesis of NIID[34–41]. To evaluate whether our gene-editing strategy could effectively target the expanded GGC repeats and reduce polyG levels, we established a HEK293 cell line stably expressing *NOTCH2NLC*-98GGC-GFP (Fig. 2A). The GFP tag (start codon removed)

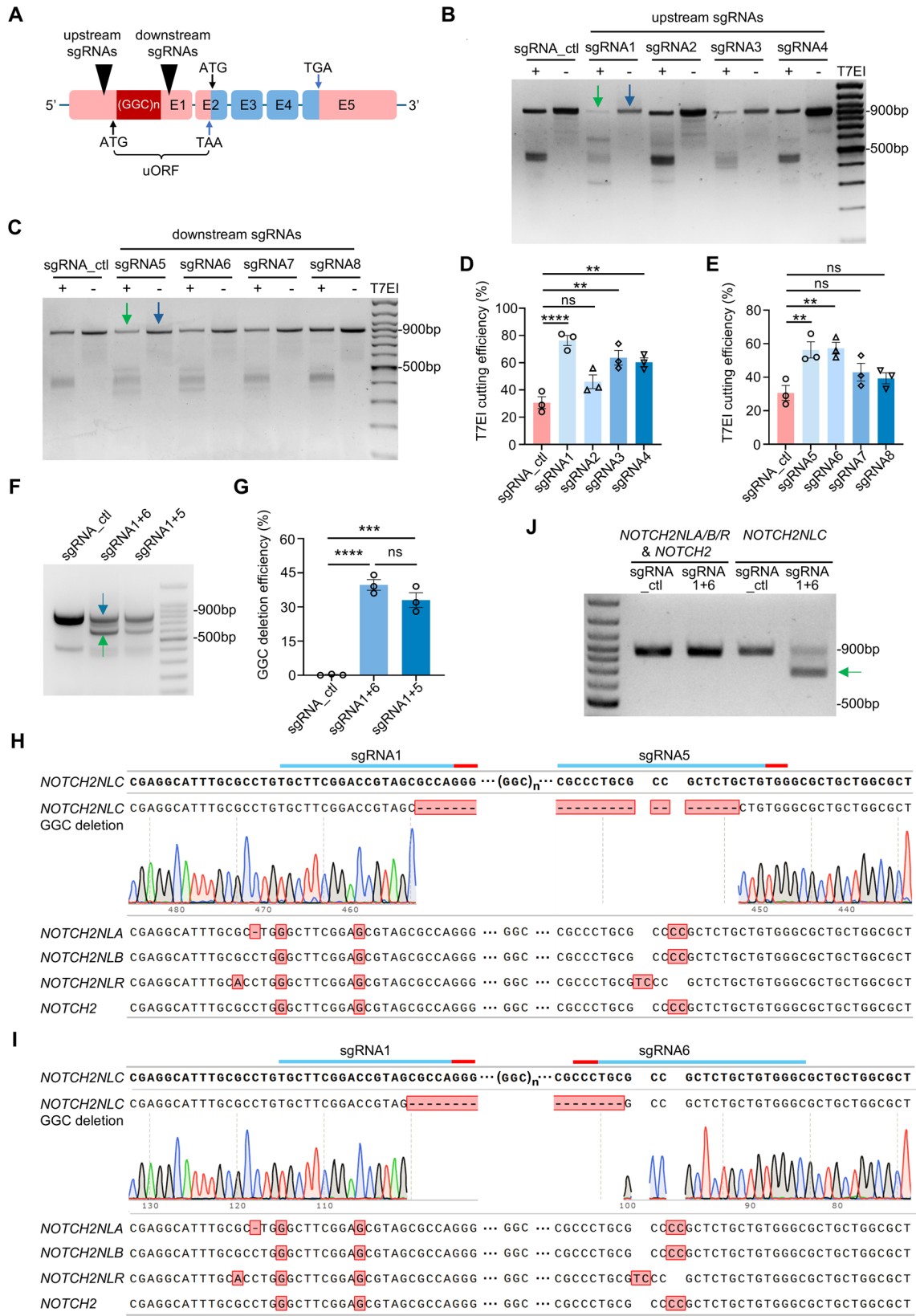

was fused to the polyG for detecting its expression. PCR analysis confirmed the expansion of GGC repeats in this cell line, and western blotting analysis showed the translation of the expanded GGC repeats into polyG-GFP (Fig. 2B, C). We then co-transfected the stable cell line with Cas9 and sgRNA combinations (1 + 6, 1 + 5) or control sgRNA to test the effect of our gene-editing strategy.

Western blot analysis demonstrated a significant reduction in the levels of both aggregated and soluble forms of polyG (by 49.0–51.7% and 67.6–78.4%, respectively) following treatment with Cas9 and *NOTCH2NLC* sgRNA combinations (Fig. 2D, E). Consistent with these findings, immunofluorescence staining results showed that both *NOTCH2NLC* sgRNAs combinations significantly reduced the number

**Fig. 1 | CRISPR/Cas9-based gene-editing efficiently and specifically targeted *NOTCH2NLC* GGC repeats in HEK293 cells. A** Schematic diagram of a CRISPR/Cas9-based gene-editing strategy targeting the GGC repeats within the upstream open reading frame (uORF) of *NOTCH2NLC*. Two mismatch-flanking regions adjacent to the GGC repeats were selected for the design of specific sgRNAs. The coding sequences and untranslated regions of *NOTCH2NLC* are indicated in blue and pink, respectively. **B–E** HEK293T cells were co-transfected with Cas9 and different sgRNAs, and the T7EI assay was performed to evaluate the targeting efficiency of sgRNAs designed for the upstream (B&D) and downstream (C&E) mis-match regions. PCR amplification using *NOTCH2NLC* specific primers yielded a ~900 bp product. The T7EI cutting efficiency (%) = [1 − (gray value of cut band)/(gray value of uncut band)] × 100%. The cut and uncut bands are indicated by green arrow and blue arrow respectively. **F**, **G** Agarose gel electrophoresis evaluated the GGC deletion efficiency of selected sgRNA combinations. The GGC deletion efficiency (%) = [(gray value of edited band)/(gray value of edited band + unedited band)] × 100%. The edited and unedited bands were indicated by green arrow and blue arrow respectively. A reduction of ~150 bp in the PCR product size was observed due to the GGC deletion. **H**, **I** TA cloning and Sanger sequencing of PCR products from the homologous regions of *NOTCH2* and *NOTCH2NL* family genes confirmed the specific deletion of GGC repeats in *NOTCH2NLC* following gene-editing. **J** Agarose gel electrophoresis evaluated the specificity of GGC deletion in *NOTCH2NLC*. The truncated PCR products (indicated by green arrow) were specifically detected in *NOTCH2NLC*, but were absent in *NOTCH2* and *NOTCH2NLA/B/R*. One-way ANOVA test with multiple comparisons. In **D**, ****$P$ < 0.0001, ns no significance, **$P$ = 0.0012 (sgRNA3 vs mock), **$P$ = 0.0026 (sgRNA4 vs mock); In **E**, **$P$ = 0.0062 (sgRNA5 vs mock), **$P$ = 0.0049 (sgRNA6 vs mock); In **G**, ****$P$ < 0.0001, ***$P$ = 0.0001. $N$ = 3 independent experiments. Data are represented as mean ± SEM. Source data are provided as a Source Data file.

(by 63.1–69.7%) of polyG-GFP aggregates compared to the control group (Fig. 2F, G). Further PCR and sequencing analyses confirmed that the decrease in polyG-GFP signal was consistent with the successful deletion of GGC repeats (Supplementary Fig. 4A–E). Collectively, all these results indicate that our gene-editing strategy targeting expanded GGC repeats effectively and significantly reduced polyG expression in vitro.

## Gene-editing of expanded GGC repeats reversed transcriptomic changes in patient-derived cell model

Human iPSCs provide a more physiologically relevant system for studying the expression and function of human-specific disease genes that are expressed at the endogenous level. We therefore established iPSCs from peripheral blood mononuclear cells (PBMCs) of a NIID patient and a sex- and age-matched health control (Supplementary Fig. 5A). Repeat-primed polymerase chain reaction (RP-PCR) and GC-rich PCR (GC-PCR) assay, performed as previously described[22], determined the GGC repeat lengths to be 113/17 repeats (patient) and 19/18 repeats (control) in the respective iPSC lines (Supplementary Fig. 5A, B). All iPSC lines maintained normal karyotypes and exhibited comparable expression of pluripotency markers (Supplementary Fig. 6A, B).

Recent studies have demonstrated a promising gene-therapy strategy involving the correction of expanded CAG repeats to normal repeats via homology directed repair (HDR), which has been successfully validated in a HD pig model[46]. However, whether this gene replacement strategy can be used in NIID models remains to be explored. To address this, Cas9 with sgRNA1 + 6 and a HDR donor template with normal GGC repeats were nucleofected into the NIID iPSC line (Supplementary Fig. 7A, B). Using fluorescence-activated cell sorting (FACS), serial dilution, and PCR-based genotyping, we successfully isolated four monoclonal lines in which the expanded GGC repeats were either deleted or repaired to normal-length repeats. The deleted or repaired GGC repeats were further confirmed by RP-PCR, GC-PCR and Sanger sequencing after TA cloning (Supplementary Fig. 5C–F and Supplementary Fig. 7C–F). Furthermore, all GGC-deletion and GGC-repair iPSC lines, retained normal pluripotency and karyotypic stability (Supplementary Fig. 6A, B).

To evaluate the specificity of our gene-editing system and investigate potential off-target effects, we selected two representative edited iPSC lines, one with GGC-deletion (line-22) and one with GGC-repair (line-37), using the unedited NIID iPSC line as a control. First, we employed deep-sequencing (deep-seq) to analysis top twenty potential off-target sites (ten for sgRNA1 and ten for sgRNA6), including the highly homologous *NOTCH2* and *NOTCH2NL* gene family (Fig. 3A, B). These candidate sites were amplified by PCR using candidates-specific primers (Supplementary data 2) and subjected to deep-seq. Bioinformatic analysis confirmed precise gene-editing in *NOTCH2NLC*, with minimal off-target activity (Fig. 3A, B and Supplementary Fig. 8A, B). Second, to broadly assess genomic integrity and potential off-target activity, we performed WGS on the same iPSC lines. Our results revealed comparable and negligible extent of single-nucleotide polymorphism (SNP), insertion/deletion (InDel), structure variants and copy number variants (CNV) between edited iPSCs and unedited iPSC (Supplementary Fig. 8C). In addition, read-depth analysis of the additional 30 predicted off-target sites also showed no significant differences between groups (Supplementary Fig. 8D, E), further supporting the specificity of our editing strategy.

Subsequently, we differentiated these edited iPSCs (both GGC-deletion and GGC-repair) along with NIID and WT iPSCs into NPCs. Immunofluorescence analysis showed similar expression of PAX6 and Nestin in GGC-deletion and GGC-repair NPCs relative to WT controls, suggesting preserved neural differentiation potential (Supplementary Fig. 9). Furthermore, no significant alterations were observed in the expression levels of *NOTCH2NLA/B/C/R* and *NOTCH2* following gene editing (Supplementary Fig. 3B).

NIID develops neuronal pathology in an age-dependent manner. Although our previous studies revealed that NPCs derived from NIID iPSCs retained differentiation capacity, they displayed significant transcriptomic alterations that may be associated with the disease[39]. To investigate whether GGC-deletion or GGC-repair could reverse the transcriptomic alterations, we performed bulk RNA sequencing. Heatmap analysis and principal component analysis (PCA) revealed that the transcriptomic profiles of GGC-deletion and GGC-repair NPCs more closely resembled those of WT NPCs than NIID NPCs (Fig. 3C, D). Compared to WT NPCs, 1983 upregulated genes and 1244 downregulated genes were identified in NIID NPCs (Fig. 3E). However, after GGC repeats deletion, 50.88% (633/1244) of downregulated genes increased and 74.74% (1482/1983) of upregulated genes decreased; whereas following GGC repeats repair, 72.59% (903/1244) of downregulated genes increased and 70.30% (1394/1983) of upregulated genes decreased (Fig. 3F–I). Gene Ontology (GO) analysis on the subset genes that exhibit reversal trends revealed significant enrichment in neuronal function after GGC-deletion and GGC-repair (Fig. 3J). These findings suggest that removing or correcting the expanded GGC repeats not only preserves the pluripotency and differentiation capacity of iPSCs but also reverses the GGC expansion-mediated transcriptomic changes in iPSCs derived cells.

## Gene-editing of expanded GGC repeats alleviated disease-related pathological and clinical phenotypes in transgenic NIID mouse model

To evaluate the in vivo efficacy of our gene-editing strategy, we leveraged a transgenic NIID mouse model, previously established in our laboratory, that ubiquitously expresses the *NOTCH2NLC*-98GGC[39]. This model faithfully recapitulates pathological hallmarks and clinical features of NIID, including widespread intranuclear inclusions, progressive neurodegeneration, motor deficits, as observed in patients with NIID[39,40]. To deliver the CRISPR/Cas9 system effectively across the blood-brain barrier, we packaged Cas9 and sgRNAs into AAV-PHP.eB, a

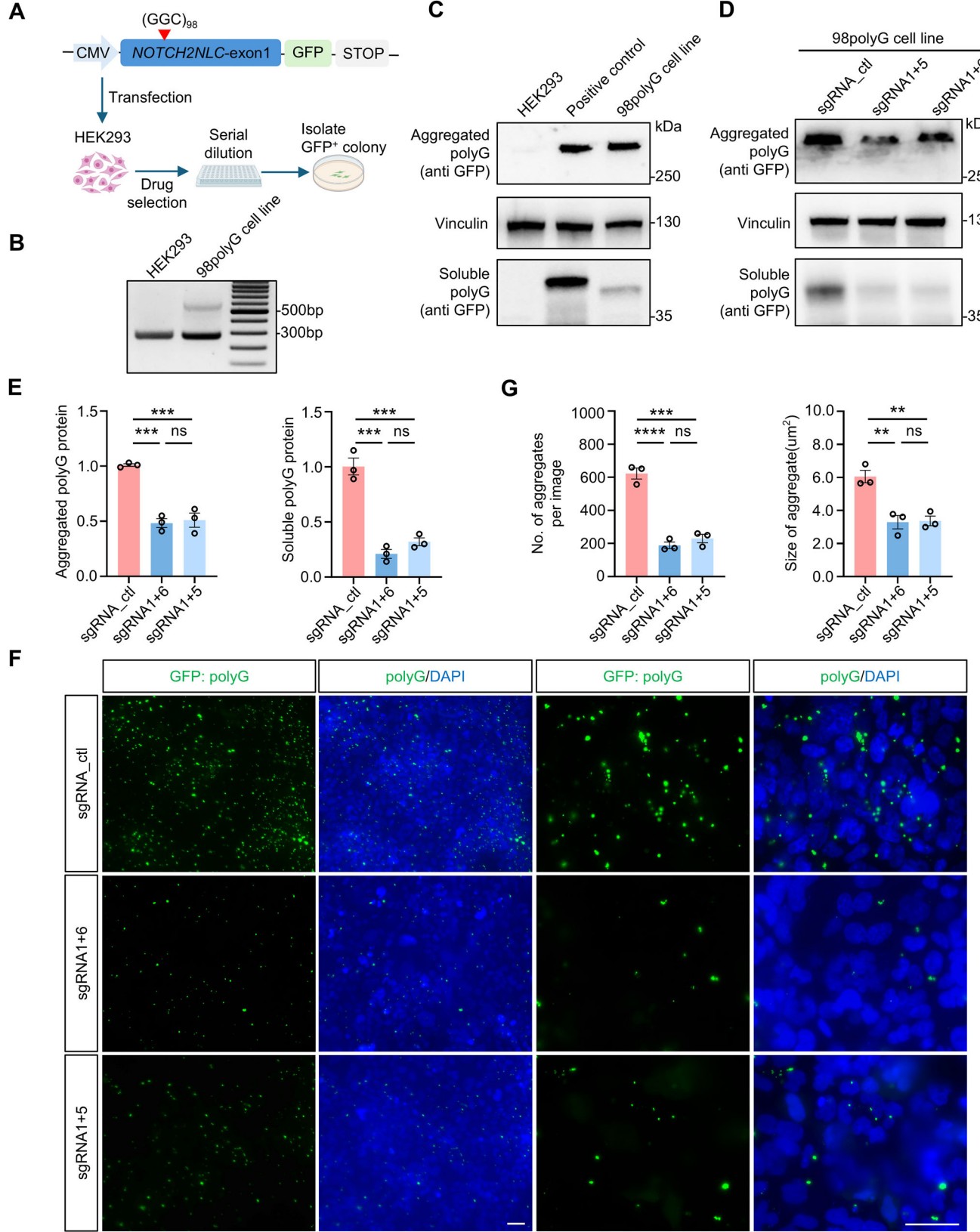

capsid known for its high transduction efficiency in the central nervous system[47].

First, we conducted the treatment in a specific brain region of NIID mice to evaluate the in vivo effectiveness of our gene-editing strategy. To minimize inter-individual variability, therapeutic AAV (Cas9 and sgRNA1 + 6) and control AAV (Cas9 and control sgRNA) were injected into the left and right striatum of the same NIID mouse, respectively. Injections were performed at postnatal day 8–10 (P8-10) with a dose of $1 \times 10^{10}$ genomic copies (GC) per side. Four weeks post-injection, immunostaining revealed a significant reduction in the number of polyG aggregates on the left striatum injected with Cas9 and sgRNA1 + 6 compared to the right side injected with Cas9 and control

**Fig. 2 | Gene-editing of expanded GGC repeats significantly reduced polyG levels in *NOTCH2NLC*-98GGC stable cell line. A** Schematic diagram of the *NOTCH2NLC*-98GGC-GFP construct and establishment of stable HEK293 cells expressing *NOTCH2NLC*-98GGC-GFP (98polyG cell line for short). GFP tag was infused with polyG proteins. **B** PCR confirmed the expansion of GGC repeats in the 98polyG cell line. The experiment was repeated at least three times with consistent results. A representative image is shown. **C** Western blotting analysis of polyG protein expression in 98polyG cell lines using anti-GFP antibody. Positive control: HEK293 cells transiently transfected with *NOTCH2NLC*−98GGC-GFP construct. Vinculin served as the loading control. The experiment was repeated at least three times with consistent results. A representative image is shown. **D, E** Western blotting analysis demonstrated that co-transfection of Cas9 with selected sgRNA combinations significantly reduced both aggregated and soluble polyG proteins levels in the 98polyG cell line. Quantification of the results is shown in (**E**). $N = 3$

independent experiments. Data are represented as mean ± SEM. One-way ANOVA test with multiple comparisons. For aggregated protein, ***$P = 0.0005$ (sgRNA1 + 5 vs sgRNA_ctl), ***$P = 0.0004$ (sgRNA1 + 6 vs sgRNA_ctl); for soluble polyG protein, ***$P = 0.0003$ (sgRNA1 + 5 vs sgRNA_ctl), ***$P = 0.0001$ (sgRNA1 + 6 vs sgRNA_ctl). **F, G** Immunostaining demonstrated that co-transfection of Cas9 with selected sgRNA combinations significantly reduced the number and the size of polyG aggregates in the 98polyG cell line. Quantification of the results is shown in (**G**). Scale bar is 40 μm. One-way ANOVA test with multiple comparisons. For number of aggregates per image, ***$P = 0.0001$, ****$P < 0.0001$; for size of aggregate, **$P = 0.004$ (sgRNA1 + 6 vs sgRNA_ctl), **$P = 0.0047$ (sgRNA1 + 5 vs sgRNA_ctl). $N = 3$ independent experiments. Data are represented as mean ± SEM. The cell selection process in (**A**) were created in BioRender. xie, n. (2025) https://BioRender.com/dysupql. Source data are provided as a Source Data file.

sgRNA (Fig. 4A). Additionally, the striatum injected with Cas9 and control sgRNA showed significant decreased levels of GFAP and IBA1, markers of astrocyte and microglial activation, respectively, and a significant increased NeuN expression (Fig. 4B–D). Consistent results were observed when compared with age-matched non-disease controls, which further confirmed the therapeutic efficacy of our editing strategy (Supplementary Fig. 10).

Subsequently, we conducted a systematic treatment via retro-orbital vein injection (ROV), a more clinically relevant injection method, in NIID mice. Previous studies have shown that NIID mice progressively develop phenotypic abnormalities starting around 40 days of age, with a median survival of ~2 months[39]. To assess the therapeutic potential of early intervention, we administered AAV ($1 × 10^{11}$ GC per mouse) at P2 and performed a comprehensive evaluation on treated NIID mice (injected with Cas9 and sgRNA1 + 6), untreated NIID mice (injected with Cas9 and control sgRNA) and control mice (injected with Cas9 and sgRNAs) at P35-50 (Fig. 5A). Immunofluorescence analysis confirmed successful transduction of AAV throughout the whole brain (Supplementary Fig. 11A). PCR performed with primers that amplify exon1 of *NOTCH2NLC* confirmed removal of expanded GGC repeats in the genomic DNA of mice post ROV injection (Supplementary Fig. 11B).

We analyzed behavioral performance in mice at P35–P40 and found that, compared to the control mice, the untreated NIID mice exhibited significantly reduced moving distance and duration in the open-field test (Fig. 5B–E), as well as markedly poorer performance in motor coordination tests such as the wire hang tests and rotarod (Fig. 5F–H). In contrast, the treated NIID mice showed significant improvement in these behavioral tests, with their performance reaching levels comparable to those of the control group (Fig. 5B–H). In addition, Kaplan-Meier survival analysis showed that the treated NIID mice exhibited significantly longer lifespans compared to the untreated NIID mice (Fig. 5I). Furthermore, western blot results demonstrated that, compared to the untreated NIID mice, the treated NIID mice exhibited significantly reduced protein levels of polyG and GFAP, alongside increased NeuN levels in the cortex, reaching levels comparable to those of non-disease control mice (Fig. 5J, K).

### Gene-editing of expanded GGC repeats alleviated disease-related transcriptomic alterations in transgenic NIID mouse model

Transcriptional dysregulation is a well-documented molecular hallmark of NIID, which is observed in patient tissues, patient-derived cells and animal models[9,37–39]. To investigate whether our CRISPR/Cas9-based strategy targeting expanded GGC repeats could reverse transcriptomic alterations associated with NIID, we performed bulk RNA-seq analysis on cortex tissues from the treated NIID mice (injected with Cas9 and sgRNA1 + 6), untreated NIID mice (injected with Cas9 and control sgRNA) and control mice (injected with Cas9 and sgRNAs). Global transcriptomic visualization via heatmap view and PCA analysis

revealed the notably altered gene expression in untreated NIID mice as compared with the control mice, and these alterations could be reversed by NIID treatment with CRISPR/Cas9 (Fig. 6A, B). For example, 1789 upregulated and 334 downregulated genes were identified in the untreated NIID mice compared to the control mice (Fig. 6C). In contrast, treated NIID mice showed markedly fewer differential expressed genes (DEGs, 171 upregulated and 134 downregulated) (Fig. 6D). Among the downregulated genes, 52.10% (174/334) were increased, while among the upregulated genes, 58.69% (1050/1789) were decreased in the treated NIID mice (Fig. 6E, F). Further analysis revealed that the reversed DEGs were enriched in disease-associated pathways, including neural function and inflammatory response, suggesting a partial restoration following GGC repeat-deletion treatment (Fig. 6G–I).

In addition, we assessed the impact of our gene editing strategy on peripheral tissue. Previous studies indicated that polyG aggregates are also present in cardiomyocytes, leading to cardiac abnormalities[40]. Western blot results showed a significant reduction (~79%) in cardiac polyG protein levels in the treated NIID mice compared to the untreated NIID mice (Supplementary Fig. 12A). Consistent with our observations in the central nervous system, transcriptomic profiling demonstrated that the treated NIID mice exhibited gene expression patterns more similar to the control mice, as evidenced by both heatmap visualization and PCA analysis (Supplementary Fig. 12B, C). Further analysis showed that the treated NIID mice had markedly fewer DEGs than the untreated NIID mice (571 vs 5372) when compared to the control mice (Supplementary Fig. 12D, E). Notably, 56.53% (277/490) - 64.03% (3126/4882) of originally dysregulated genes displayed reversal trends following treatment (Supplementary Fig. 12F, G), with these genes being significantly enriched in mitochondrial function and cardiac contraction pathways (Supplementary Fig. 12H). Taken together, analysis of NIID mice suggests that deletion of expanded GGC repeats effectively rescues transcriptomic dysregulation across both central nervous system and peripheral tissue.

## Discussion

The GGC repeat expansion in the *NOTCH2NLC* gene is not only the genetic cause of NIID, but has also been implicated in a broad spectrum of neurological disorders, such as essential tremor[48–51], amyotrophic lateral sclerosis[26,52,53], cerebral small vascular disease[54,55], multiple system atrophy[30], leukoencephalopathy[29,56], Alzheimer's disease[31,57], and Parkinson's disease[27,58], inherited peripheral neuropathy[59,60] and oculopharyngodistal myopathy type 3[32,61]. Collectively, these conditions are referred to as "*NOTCH2NLC*-related repeat expansion disorders (NREDs)"[16,20,22,48,62,63]. Mechanistically, expanded GGC repeats are mainly translated into polyG peptides, which constitute a major driver of intranuclear inclusion formation and cellular dysfunction[35–41]. Although administration of compounds or expression of downstream effector gene have been shown to partially ameliorate disease phenotypes in animal models[37,41,64], no therapy with broad and durable efficacy has yet been established.

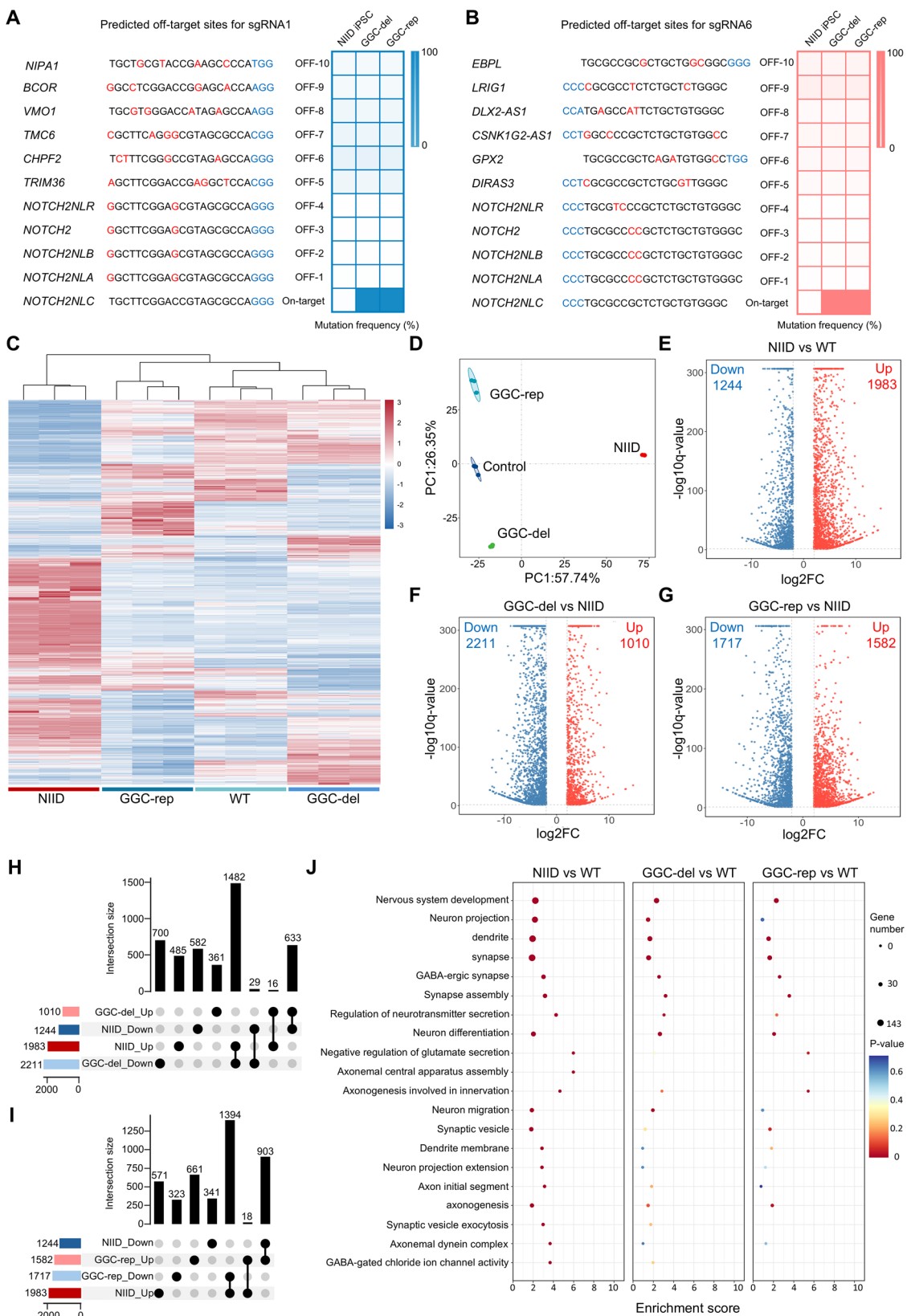

In this study, we developed a CRISPR/Cas9-based gene-editing strategy that precisely targets and excises the expanded GGC repeats in the *NOTCH2NLC* gene. This approach markedly reduced polyG aggregates in the 98G stable cell line, patient-derived iPSCs and NPCs, as well as in the transgenic NIID mouse model. Notably, in the mouse model, editing of the GGC expansion significantly ameliorated

neurodegenerative phenotypes, including the restoration of neuronal proteins, attenuation of gliosis, behavioral improvement, and extended survival. At the molecular level, removal of the expanded repeats corrected transcriptional dysregulation in both NPCs and mouse brain tissues, particularly in pathways related to neuronal function and inflammatory response. In addition, owing to the ROV administration

**Fig. 3 | Gene-editing of expanded GGC repeats reversed molecular changes in patient-derived cell model.** Deep-seq analysis revealed minimal off-target activity for sgRNA1 (**A**) and sgRNA6 (**B**) at the top 20 predicted off-target sites in both GGC-deleted and GGC-repaired iPSC colonies. Multiplex PCR-amplified potential off-target sites and target loci were deep sequenced, with off-target activity calculated as (mutated reads/total reads) *100% at each site. The heatmap analysis (**C**) and principal component analysis (PCA) (**D**) of RNA-seq data showing transcriptomic similarity among indicated groups. Transcriptomic profiles of GGC-deletion and GGC-repair NPCs more closely resembled those of WT NPCs than untreated NIID NPCs. **E–G** Volcano plots showing significantly up-regulated (depicted in red) and down-regulated (depicted in blue) DEGs among different groups (*N* = 3 biological replicates per group). The complete lists of DEGs are available in Supplementary Data 3. **H, I** Upset plot illustrating the overlap and unique DEGs among the different comparison groups shown in (**E–G**). The dark blue and dark red rectangles

indicated genes down-regulated and up-regulated in the unedited NIID NPC, respectively. The light blue and pink rectangles indicated genes down-regulated and up-regulated in the edited NIID NPC, respectively. Among 1244 downregulated genes in NIID NPCs (relative to WT NPCs), 633 and 903 genes were upregulated after GGC-deletion (**H**) and GGC-repair (**I**); while among 1983 upregulated genes in NIID NPCs, 1482 genes and 1394 genes were downregulated after GGC-deletion (**H**) and GGC-repair (**I**). **J** Gene Ontology (GO) analysis on the subset genes exhibiting reversal trends revealed significant enrichment in neuronal function after GGC-deletion and GGC-repair. For all panels, differential expression analysis was performed using DESeq2 (two-sided Wald test), with DEGs defined as |log2(fold change)| ≥ 2 and FDR < 0.05 (Benjamini-Hochberg method). GO enrichment was assessed using a one-sided hypergeometric test with FDR adjustment. Source data are provided as a Source Data file.

of the PHP.eB capsid AAV, our gene-editing strategy partially alleviated molecular abnormalities in peripheral tissue of the mouse model. Considering the multisystem involvement of NIID[19,20], further optimization of delivery strategies may enhance the therapeutic efficacy of this approach. Together, these findings highlight the central pathogenic role of GGC repeats in NIID and support permanent correction at DNA-level as a promising therapeutic avenue.

Given the high sequence similarity among *NOTCH2NLC*, its *NOTCH2NL* paralogs, and *NOTCH2*, we carefully designed sgRNAs targeting regions with minimal sequence homology and implemented a dual-sgRNA strategy to ensure editing precision. This approach achieved highly specific targeting of the *NOTCH2NLC* locus with minimal off-target activity across HEK293 cell line, 98G stable cell line, patient-derived iPSCs and NPCs. The expression of homologous genes remained unaltered (Supplementary Fig. 3), and edited iPSCs preserved normal karyotype and differentiation potential (Supplementary Figs. 6 and 9). Moreover, by providing a repair template with normal repeat length, we achieved HDR-mediated correction of the expanded GGC repeats to normal repeat length in dividing cells like iPSCs. Although HDR-based repair remains limited in postmitotic neurons, it is more feasible in proliferative cells of both the nervous system and peripheral tissues. Refining repeat-targeted repair strategies may further improve the therapeutic outcomes of this approach.

In addition to NREDs, expansions of GGC or CCG repeats in the 5' UTR of various genes have been implicated in an expanding spectrum of neurological and muscular disorders, such as FXS, FXTAS[65], FRAXE intellectual disability[66,67], movement disorders[68,69], oculopharyngeal-related myopathies[23,70,71]. Although previous studies in FXS have explored CRISPR/Cas9-mediated editing of the *FMR1* CGG repeats in cell line and iPSC[72,73], these studies were restricted to in vitro settings and did not rigorously assess potential off-target effects. In contrast, our work demonstrates the feasibility and efficacy of this strategy in both cellular and animal models of NIID, while also addressing the additional challenge posed by high sequence homology. Importantly, our study provides the preclinical proof demonstrating the efficacy of gene editing for *NOTCH2NLC* GGC repeat expansion in NIID. Furthermore, it offers a conceptual framework for applying CRISPR/Cas9-mediated repeat excision to a broader class of GGC-driven REDs.

Despite the robust efficacy and safety of our gene-editing strategy demonstrated in both in vitro and in vivo NIID models, several limitations remain. First, the transgenic NIID mouse model used in this study reproduces key pathological hallmarks of the human disorder, providing a valuable platform for therapeutic assessment. However, its rapid disease progression and short lifespan contrast sharply with the late-onset, slowly progressive nature of NIID in patients[16,74], likely reflecting species-specific differences and the absence of native regulatory mechanisms for the human-specific *NOTCH2NLC* gene. In addition, the model may not recapitulate the typical MRI signatures observed in NIID patients[16,18], such as corticomedullary hyperintensity, which may be attributed to the relatively simple white matter

architecture and lower glial density in mice[75]. Another limitation is the lack of long-term safety evaluation. Although we combined targeted deep-seq with WGS, and RNA-seq to assess safety in several human cell models and mouse model, these analyses were limited in duration and sensitivity. Incorporating unbiased, genome-wide approaches such as GUIDE-seq[76] in future work will be essential for comprehensive off-target assessment prior to clinical translation. The development of large-animal NIID models, such as pigs or non-human primates, whose brain architecture and immune responses more closely resemble humans, will be indispensable for assessing the long-term safety, delivery efficiency, and immune compatibility of this CRISPR/Cas9-based therapy in a more clinically relevant context.

In conclusion, our study provides compelling evidence that precise removal of expanded GGC repeats in the *NOTCH2NLC* gene using CRISPR/Cas9 represents a safe and effective therapeutic strategy for NIID, laying the groundwork for future clinical translation.

## Methods

### Human participants
One NIID patient and one health control from mainland China were recruited in this study. Each participant underwent a neurological examination performed by at least two experienced neurologists. The patient was a 57-year-old male with a history of bradykinesia and hand tremors for over 10 years underwent physical examination, which revealed increased muscle tone in all four extremities and action tremors in the upper extremities. Cognitive assessment yielded an MMSE score of 28 and a MoCA score of 27. The health control was a 50 year-old male. Peripheral blood samples were collected from all participants after obtaining written informed consent. These samples were subsequently reprogrammed into iPSCs for research purposes.

### GGC repeat size determination
The GGC repeat size within *NOTCH2NLC* was verified and determined using RP-PCR and GC-PCR, as previously described[22]. RP-PCR was performed with a FAM-labeled gene specific primer (CCTCAGCCC-GATACTCACCAT) and a repeat-containing primer [TACCAATACG CATCCCGCGATTTGTCTTA(CGG)5]; GC-PCR was performed with a FAM-labeled forward primer (AGCGCCAGGGCCTGAGCCTTTGAAG-CAG) and a reverse primer (TCGCCCCAGGTGGCAGCCCCGGGCGC CGCGGAC). Amplified products were analyzed by capillary electrophoresis (3500xL Genetic Analyzer, Applied Biosystems) with allele sizing based on the GeneScan 1000 ROX Size Standard.

### Animals
The transgenic NIID mouse model (C57BL/6J background) that ubiquitously expresses *NOTCH2NLC*-98GGC was generated previously, and no significant differences in pathological or behavioral phenotypes between males and females[39]. Heterozygous NIID mice and the littermate controls were used in this study. Both sexes were used for

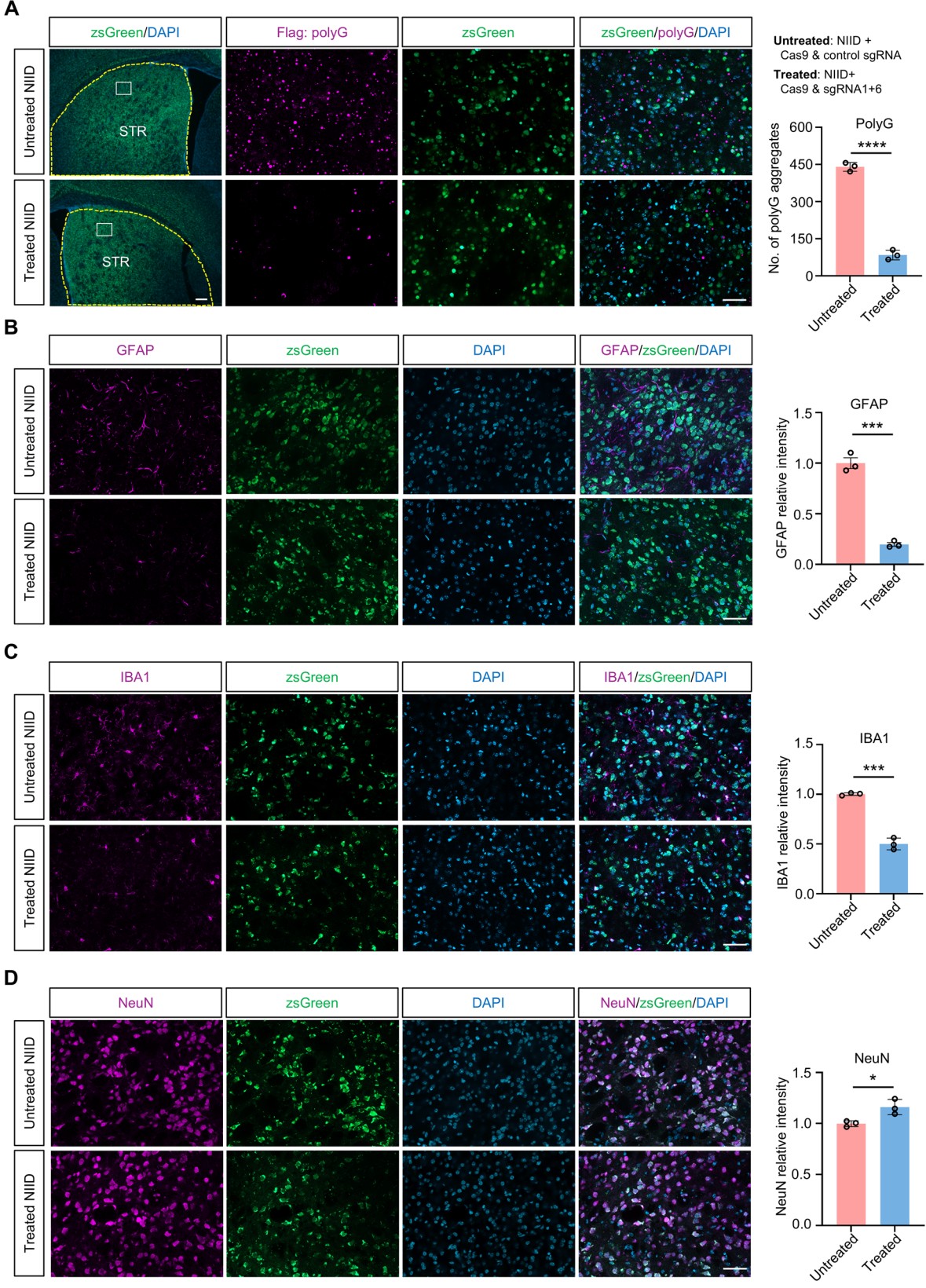

stereotaxic injection, ROV, and behavioral tests. All mice were bred and maintained on a 12:12 h light/dark cycle (lights off at 7 p.m.) in the animal facility under specific pathogen-free conditions in the animal facility of Xiangya Hospital, Central South University. The temperature was maintained at 22±1 °C with relative humidity (30–70%).

**Derivation of iPSC and differentiation to NPCs**

PBMCs from NIID patient and healthy control were reprogrammed into iPSCs using CytoTune-iPS 2.0 Sendai virus reprogramming kit according to the manufacturer's protocol. The iPSCs were cultured on vitronectin-coated plates in StemPro-34 SFM (Gibco) medium. After 3–4 weeks, the medium was replaced by Essential 8 (Gibco) when iPSC

**Fig. 4 | Localized gene-editing of expanded GGC repeats reduced polyG level and rescued neuropathology in transgenic NIID mouse model.**
**A** Immunofluorescence analysis using antibody against Flag revealed a significant reduction of polyG aggregates in the striatum injected with AAV-Cas9 and AAV-sgRNA1 + 6 (treated), compared to the contralateral side injected with AAV-Cas9 and AAV-control sgRNA (untreated). Immunofluorescence analysis using antibodies against GFAP, IBA1, and NeuN demonstrated reduced GFAP (**B**) and IBA1 (**C**) expression alongside increased NeuN (**D**) levels in the treated striatum, compared to the contralateral untreated side. ZsGreen fluorescence (reporter encoded in the AAV-sgRNA vector) confirmed efficient AAV transduction in targeted regions. Scale bar is 200 μm and 50 μm for low- and high-magnification images, respectively. ****$P < 0.0001$ (polyG), ***$P = 0.0001$ (GFAP & IBA1), *$P = 0.0245$ (NeuN), two-tailed $t$-test. $N = 3$ mice per group. Data are represented as mean ± SEM. Source data are provided as a Source Data file.

colonies were established. The iPSC lines were validated by karyotyping to confirm chromosomal constitution, and the specific GGC repeat expansion within the patient-derived line was verified by RP-PCR and GC-PCR. IPSCs were differentiated toward the neural progenitor lineage as previously reported[39]. In brief, NPCs were derived from iPSCs using the monolayer culture protocol with STEMdiffTM Neural Induction Medium (Stemcell technologies). The generated NPCs were maintained in neural progenitor medium (STEMCELL Technologies) on Matrigel-coated and used for subsequent experiments. All cell lines were negative for mycoplasma.

## Plasmids and viruses

The *NOTCH2NLC*-98GGC, Cas9, and sgRNAs plasmids were constructed in previous studies[39,45,46]. The sequences of sgRNA are listed in the Supplementary Data 1. The non-targeting control sgRNA was verified previously[45]. Cas9 and sgRNA vectors were sent to PackGene Biotech (Guangzhou, China) for viral packaging and production. All the viral vectors were packaged into the PHP.eB serotype. The titers of the viruses used for stereotactic injection and retro-orbital injection were $1 \times 10^{13}$ GC/mL and $5 \times 10^{13}$ GC/mL, respectively.

## Antibodies

Primary antibodies used in this study include the following: Flag (Sigma-Aldrich, F1804, 1:1000), vinculin (Sigma-Aldrich, V9131, 1:1000), NeuN (Cell Signaling Technology, 24307s, 1:1000), GFAP (Cell Signaling Technology, 3670S, 1:1000), IBA1 (Wako, 019-19741, 1:1000), GFP (Invitrogen, A-11122, 1:2000), zsGreen (Sangon, D199984, 1:2000), OCT4 (Proteintech, 60242-1-Ig, 1:200), SOX2 (Proteintech, 11064-1-AP, 1:1000), PAX6 (Abcam, AB195045, 1:350), Nestin (CST, 33475S, 1:2000). Fluorescent secondary antibodies (1:200) used were donkey anti-rabbit and donkey anti-mouse antibodies conjugated with Alexa Fluor 488 or 594 from Jackson ImmunoResearch. HRP-conjugated secondary antibodies (1:10000) were donkey anti-rabbit and donkey anti-mouse from Jackson ImmunoResearch.

## Cell culture and transfection

HEK293 cells were cultured in Dulbecco's modified Eagle's medium supplemented with 10% fetal bovine serum and penicillin/streptomycin (100 u/ml) at 37 °C with 5% $CO_2$. For the transfection procedure, the cells were seeded onto 12-well plates at a density of ~200,000 cells per well. After 24 h, the cells were transfected with plasmids using Lipofectamine 3000 according to the manufacturer's instructions. 48–72 h after the transfection, the cells were harvested for subsequent experiments. HEK293 cells (Procell, CL-0001) were authenticated by STR profiling and were tested negative for mycoplasma contamination.

## Stable cell line establishment

200,000 HEK293 cells were co-transfected with a total of 1 μg *NOTCH2NLC*-98GGC-GFP plasmid. DMEM was supplemented with G418 (600 μg/mL) for selection. 48 h after transfection, the cells were treated with trypsin and then reseeded in 96-well plates containing selective medium. A serial dilution approach was adopted during this process. The selective medium was refreshed every 3–4 days to eliminate non-transfected cells while allowing resistant colonies to proliferate. Once visible G418-resistant foci emerged, they were isolated

and expanded sequentially into 24-well plates and later into 6-well plates for validation of stable integration.

## Nucleofection and monoclonal cell isolation

One day before the passage day, iPSCs were treated with 10 μM Y-27632 overnight. On the passage day, $1 \times 10^6$ iPSCs resuspended in human stem cell nucleofector solution I (Lonza, Basel, Switzerland) were nucleofected with 2 μg of plasmids (Cas9: sgRNA = 1:1) using program B16 in the nucleofector IIb device (Lonza). After nucleofection, the cells were transferred to a freshly Matrigel-coated plate and cultured in mTeSR1 medium containing 10 μM Y-27632 for 1 day, followed by regular mTeSR1 medium. 24–48 h after nucleofection, the iPSCs sorted on a Becton Dickinson FACS Aria II cell sorter were seeded onto a 24-well plate layered with mouse embryonic fibroblasts at a density of 100 cells per well. iPSC colonies emerged between 10-and 12-days post-sorting. These colonies were transferred to freshly Matrigel-coated plates for genotyping. A total of ninety clones were analyzed and four monoclonal lines, in which the expanded GGC repeats were either deleted or repaired to normal-length repeats, were selected for subsequent experiments.

## T7EI assay

T7 Endonuclease I assay was performed as described previously[77]. Genomic DNA was extracted from HEK293 cells using a commercial DNA isolation kit (Magen). To amplify the target regions flanking the CRISPR-Cas9 editing site, PCR was performed with LA Taq (Takara) using primers specific to amplify *NOTCH2NLC* or *NOTCH2/NOTCH2NLA/B/R*. The T7EI primers are listed in the Supplementary Data 2. The PCR products were purified and subjected to heteroduplex formation by denaturation and gradual reannealing to allow mismatched base pairing between the wild-type and mutant alleles. The heteroduplex DNA was digested with T7 Endonuclease I (New England Biolabs) at 37 °C for 5 min, followed by enzyme inactivation. The digested fragments were separated on a 1.5% agarose gel, and the band intensities were quantified using the ImageJ software to calculate the cutting frequency.

## TA cloning

Genomic DNA region encompassing GGC repeats was amplified using primers that amplify *NOTCH2NLA/B/C/R* and *NOTCH2* simultaneously. The TA clone primers are listed in the Supplementary Data 2. PCR products were purified using a commercial gel extraction kit (Magen). TA cloning was performed using a commercial SuperTOPO TA clone kit (Vazyme). The ligation mixture was transformed into chemically competent *E. coli* DH5α cells, followed by plating onto LB agar containing ampicillin at 37 °C overnight for colony selection. The next day, colonies were picked and cultured in LB broth. Plasmid DNA was isolated and sequenced with M13 forward primer. Sanger sequencing results were aligned and analyzed using the SnapGene software.

## Reverse transcriptase PCR and quantitative real-time PCR

Total RNA was extracted from cell pellets using a TRIzol reagent (Invitrogen) following the manufacturer's instructions. cDNA was synthesized from 1 μg of total RNA using the RevertAid First-Strand cDNA Synthesis Kit (Thermo Fisher Scientific) and oligo dT primers. Quantitative real-time PCR was performed and analyzed by applying

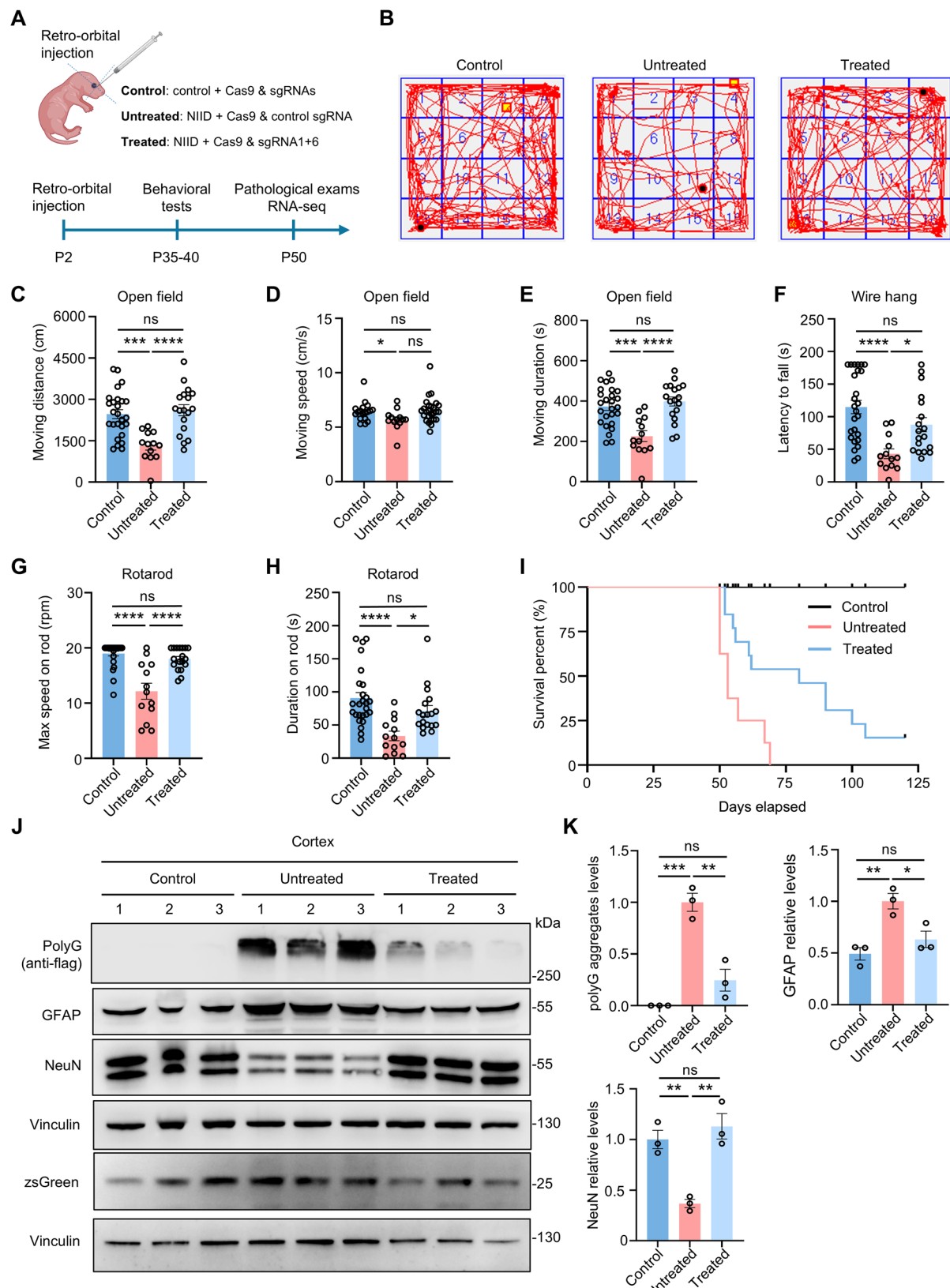

the comparative cycle threshold method. The primers used for the quantitative real-time PCR are listed in the Supplementary Data 2.

### Western blot
Mouse tissues or cell pellets were lysed in ice-cold RIPA buffer (50 mM Tris, pH 8.0, 150 mM NaCl, 1 mM EDTA pH 8.0, 1 mM EGTA pH 8.0, 0.1% SDS, 0.5% DOC, and 1% Triton X-100) containing protease inhibitor. Protein concentration was determined with the BCA Protein Assay Kit (Thermo Scientific). Protein samples were separated by SDS-polyacrylamide gel electrophoresis and transferred to a poly-vinylidene fluoride membrane (Millipore), which was further blocked with 5% milk/PBS for 1 h at room temperature. The blot was incubated

**Fig. 5 | Systematic gene-editing of expanded GGC repeats ameliorated behavioral and pathological phenotypes in transgenic NIID mouse model.**
**A** Schematic diagram of retro-orbital vein injection (ROV) in neonatal mice and the timeline for behavioral tests, pathological exams and RNA-seq. Three groups of mice were used. Treated group: NIID mice injected with AAV-Cas9 and AAV-sgRNA1 + 6; Untreated group: NIID mice injected with AAV-Cas9 and AAV-control sgRNA; Control group: littermate control mice injected with AAV-Cas9 and AAV-sgRNA. Motor functions of mice in different groups were evaluated using open field test (**B**–**E**), wire hang (**F**), and rotarod tests (**G**, **H**). Treated NIID, $N = 18$; untreated NIID, $N = 13$; control, $N = 26$. **I** Kaplan-Meier survival curve of mice with or without gene therapy (treated NIID, $N = 13$; untreated NIID, $N = 8$; control, $N = 20$). **J**, **K** Western blotting of cortical tissue revealed that treated NIID mice exhibited reduced polyG and GFAP expression alongside increased NeuN levels, compared to untreated NIID mice. The GFAP and NeuN levels of treated NIID mice were restored to levels comparable to those in the control group. Successful AAV transduction was confirmed by zsGreen expression in all experimental animals. Vinculin served as the loading control. $N = 3$ mice per group. One-way ANOVA test with multiple comparisons. In **C**, ***$P = 0.0002$, ****$P < 0.0001$. In **D**, *$P = 0.0475$. In **E**, ***$P = 0.0001$, ****$P < 0.0001$. In **F**, *$P = 0.0263$, ****$P < 0.0001$. In **G**, ****$P < 0.0001$. In **H**, *$P = 0.0189$, ****$P < 0.0001$. In **K**, for polyG aggregates levels, **$P = 0.0013$, ***$P = 0.0003$; for GFAP levels, *$P = 0.0255$, **$P = 0.0059$; for NeuN levels, **$P = 0.0027$ (treated vs untreated), **$P = 0.0068$ (untreated vs control). Data are represented as mean ± SEM. The neonatal mouse in Figure 5A was created in BioRender. xie, n. (2025) https://BioRender.com/dysupql. Source data are provided as a Source Data file.

---

with primary antibodies overnight at 4 °C. After three washes in 0.05% PBST, the blot was incubated with HRP-conjugated secondary antibodies for 1 h at room temperature. After three washes in 0.05% PBST, protein bands were developed using an ECL Prime Chemiluminescence kit (GE Healthcare) on a Bio-Rad developing and imaging platform. Protein band intensities were quantified using ImageJ to calculate expression levels relative to internal controls.

### Immunofluorescent staining
Mice were anesthetized and perfused intracardially with 0.9% saline solution, followed by 4% paraformaldehyde in 0.1 M phosphate buffer at pH 7.2. The isolated mouse brains were dehydrated in 30% sucrose at 4 °C and sectioned at a thickness of 30 μm for the subsequent immunofluorescence study. Brain sections were blocked with a solution of 3% bovine serum albumin (BSA) in 0.3% Triton X-100/PBS for 60 min, followed by incubation with primary antibodies overnight at 4 °C. After incubation with fluorescent secondary antibodies and 4′,6-diamidino-2-phenylindole (DAPI), the brain sections were mounted onto coated glass slides and examined using a Zeiss microscope (Apotome 3).

For immunostaining of iPSCs and NPCs, cells grown on coverslips were fixed with 4% paraformaldehyde for 10 min and permeabilized with 0.1% Triton X-100 for 15 min at room temperature. Throughout the entire staining process, a solution consisting of 2% normal goat serum and 3% BSA in PBS was used for blocking. Incubation with primary antibodies was carried out overnight at 4 °C and followed by fluorescent secondary antibody incubation for 45 min at room temperature. Images were captured using a Zeiss microscope (Apotome 3). The number of polyG aggregates and immunofluorescent intensities of GFAP, IBA1, and NeuN were quantified using ImageJ software.

### Stereotaxic injection and retro-orbital vein injection
During the striatal stereotaxic injection, P8-10 mice were anesthetized via inhalation of 1.5% isoflurane and then firmly positioned in a stereotaxic instrument. AAV-sgRNA and AAV-Cas9 were mixed at a ratio of 1:1. 1 μl of the mixed viruses ($1 \times 10^{10}$ GC) was injected into one side of the mouse striatum according to the following coordinates adjusted to the flat skull position: 0.39 mm rostral to the bregma, 1.4 mm lateral to the midline, and 2.5 mm ventral from the dural surface over a period of 5 min. An RWD stereotaxic instrument (69100) and a Hamilton microsyringe (1700 series, 10 μl) were used to deliver the virus at a speed of 100 nl per min. After the surgery, the mice were placed on a heated blanket to recover. All stereotaxic surgeries were completed successfully, with incisions healing well and no adverse events observed.

ROV was conducted following a previously established protocol[78]. Briefly, P1-P2 neonatal mice were cryo-anesthetized on ice for 3 min. The viruses expressing sgRNA and Cas9 were mixed at a ratio of 1:1 and 2 μl of the mixed viruses ($1 \times 10^{11}$ GC) were injected into each pup. To perform the injection, place the pup in left lateral recumbency with its head facing to the right. Gently hold the pup's head with the thumb and forefinger of the non-dominant hand. Identify the retro-orbital sinus region and insert ~1/3 of the length of a 10 μL needle, with the bevel facing down, at a 45° angle into the inner canthus. After the injection, the pups were placed on a heated blanket to recover. Among sixty-four injected pups, fewer than ten died due to poor rewarming after hypothermic anesthesia or cannibalization by the dam, while the remaining mice survived and underwent subsequent experiments. Injections were performed in a blinded manner, as the genotypes were unknown at P1-2. For each litter, either therapeutic or control AAV was administered uniformly. After genotyping, transgenic mice were assigned to either treated (with therapeutic AAV) or untreated (with control AAV) NIID groups, and all control littermates were assigned to the control group.

### Mouse behavioral analysis
Behavioral tests were performed as previously described[39]. At least 13 mice per genotype were included. The open field test was performed in a Perspex box (40 cm in length, 40 cm in width, and 31 cm in height) with the floor divided into 16 equal-sized squares. During the 10-min test period, the movement of the mice was recorded on video and analyzed by an automated behavior analysis system. The total moving duration and distance were then compared between different groups. In the wire-hang test, mice were placed on the wire lid of a mouse cage, which was then inverted. During the 3-min test period, the latency to fall was recorded. Each mouse underwent the test twice, and the average hang time was compared across different groups. For the rotarod test, mice were trained for 3 min on the rotarod, with three trials per day for three consecutive days. On the testing day, each mouse underwent an accelerating-speed test (from 5 to 20 rpm) for 3 min, and this test was repeated twice. The intertrial interval was 10 min, and the latency to fall was recorded for each trial. The average time the mice spent on the rotarod was compared among different groups. The survival time of each mouse was monitored and recorded daily.

### Deep-sequencing analysis
Potential off-target sites were predicted using the online tool CRISPOR. We generated a PCR library for next-generation sequencing (NGS). We first used specific primers to amplify genomic DNA containing the targeting region with LA Taq (Takara). The second PCR was then performed with the same primers containing Illumina forward and reverse adapters to generate the PCR library for NGS. The final 50 μl PCR products were purified using a commercial gel extraction kit (Magen) and qualified with an Agilent Bioanalyzer 2100. The PCR products were then sent to IGE Biotechnology (Guangzhou) for sequencing. The NGS data were analyzed by CRISPResso2[79]. The top 20 potential off-target sites and primer sequences are listed in the Supplementary Data 2.

### RNA-seq and analysis
Mice cortex and heart, and human NPCs were used for RNA extraction. An amount of 1 μg of total RNA per sample was processed for library preparation using VAHTS Universal V10 RNA-seq Library Prep Kit following the manufacturer's recommendations. Libraries were sequenced on an Illumina NovaSeq X Plus (mice tissues) or Illumina Novaseq 6000 (human NPCs) platform. Raw reads were processed by

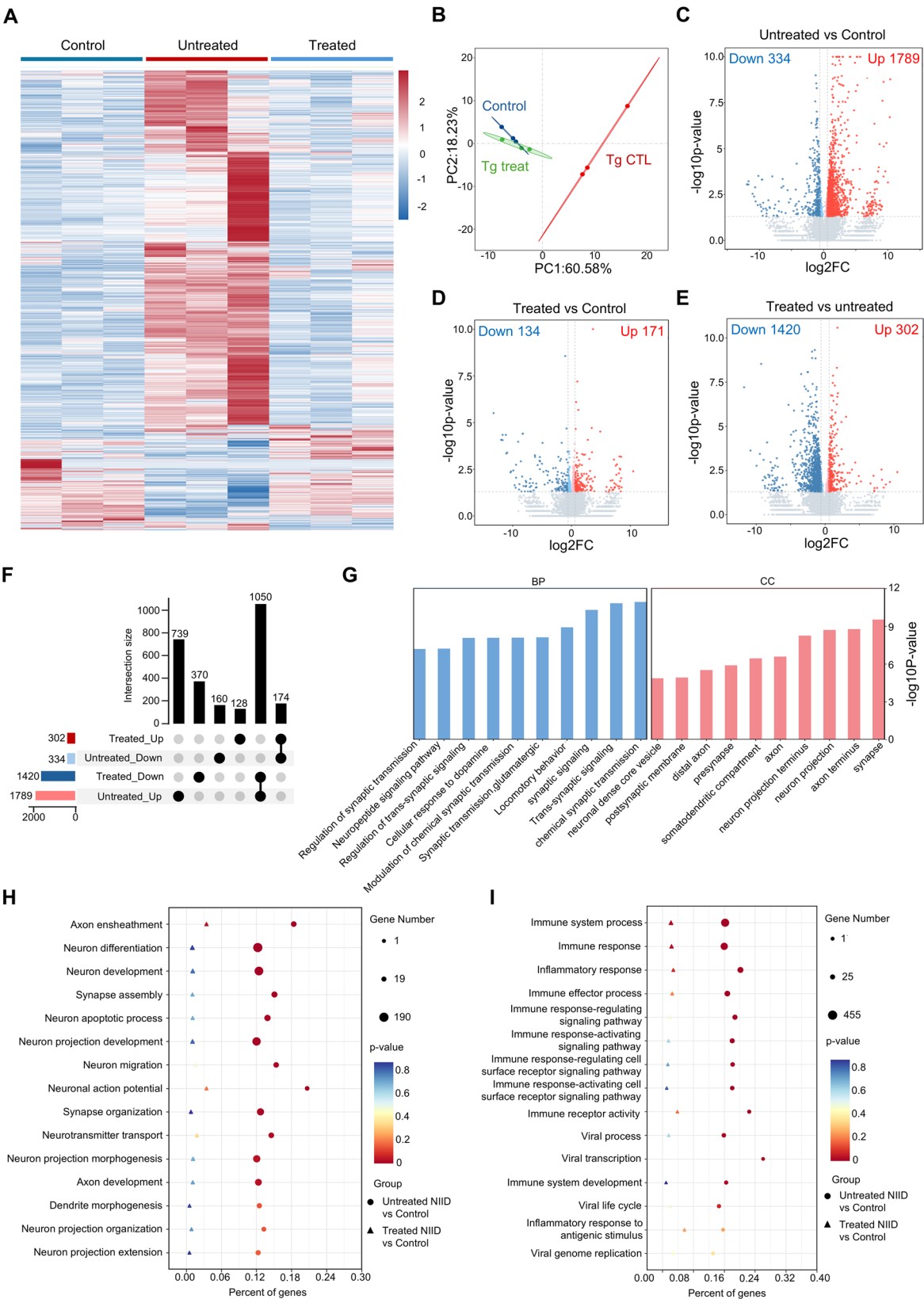

fastp to get clean reads. Paired-end clean reads were mapped to GRCm38 or GRCh38 by HISAT2 and assembled by StringTie. Quantification of gene expression levels was estimated by TPM (mice tissues) or FPKM (human NPCs). Heatmap and PCA analysis were performed using R[80]. Differential expression analysis was performed using DESeq2. GO enrichment analysis of DEGs was performed using R[81].

## Whole-genome sequencing

WGS was performed by Novogene, Co., LTD. A total amount of 0.2 µg DNA per sample was used as input material for the DNA library preparations. Sequencing library was generated using NEB Next® Ultra™ DNA Library Prep Kit for Illumina (NEB, USA) following manufacturer's recommendations and index codes were added to each sample. Next,

**Fig. 6 | Gene-editing of expanded GGC repeats alleviated disease-related transcriptomic alterations in transgenic NIID mouse model.** The heatmap (**A**) and PCA (**B**) analysis of RNA-seq data on cortex showing transcriptomic similarity among indicated groups. Transcriptomic profiles of treated NIID mice more closely resembled those of control mice than untreated NIID mice ($N = 3$ biological replicates per group). **C–E** Volcano plots showing up-regulated (depicted in red) and down-regulated (depicted in blue) DEGs among different groups. The complete lists of DEGs are available in Supplementary Data 4. **F** Upset plot showing the overlap and unique DEGs among the different comparison groups shown in (**C–E**). Among 334 down-regulated genes in untreated NIID mice (relative to control mice), 174 genes were upregulated after treatment; while among 1789 upregulated genes in untreated NIID mice, 1050 genes were downregulated after treatment. **G** GO enrichment analysis revealed that the reversed DEGs were enriched in neural-related pathways. **H, I** GO enrichment analysis comparing untreated NIID-vs-control and treated NIID-vs-control revealed that neural function and inflammatory response related pathway was partially ameliorated in treated NIID mice. For all panels, differential expression analysis was performed using DESeq2 (two-sided Wald test), with DEGs defined as $|\log2(\text{foldchange})| \geq 0.585$ and $P < 0.05$. GO enrichment was assessed using a one-sided hypergeometric test with FDR adjustment (Benjamini-Hochberg method). Source data are provided as a Source Data file.

the DNA libraries were sequenced on Illumina platform and 150 bp paired-end reads were generated. The effective sequencing data was aligned with the reference sequence through BWA[82] software. Samtools[83] and bcftools were used to extract all potential polymorphic SNPs and InDel sites from the entire genome with the mapped reads. BreakDancer[84] software was used to detect insertion, deletion, inversion, intra-chromosomal translocation and inter-chromosomal translocation mutations, based on the reference genome mapping results and the detected insert size. CNVnator[85] was used to detect CNVs of potential deletions and duplications. Cas-OFFinder[86] was used to predict off-targets across the entire genome. Additional 30 off-target loci in the experimental group were compared with the control group. The relative depths were calculated by taking the number of mapped reads on these sites into the whole genome of the pointer reads.

## Statistical analysis and reproducibility
All statistical analyses were performed with either two-tailed Student's *t* test or the one-way ANOVA test with multiple comparisons using GraphPad Prism 9. All quantitative data were presented as mean ± SEM. A *P* value of less than 0.05 was considered statistically significant. Sample sizes were chosen based on established standards in the field to ensure robustness. No statistical method was used for predetermination, and no data were excluded. For in vivo studies (e.g., behavior, survival; $N \geq 8$ per group), littermates were randomly allocated and investigators were blinded during procedures and assessment. For in vitro assays ($N \geq 3$ independent replicates), while group allocation was defined by experimental conditions, sample processing and analysis order were randomized. Quantitative analysis for in vitro data was performed with blinding to group identity.

## Ethics statement
All experimental protocols involving human participants and animals were conducted in compliance with all relevant ethical regulations. The study involving human participants was reviewed and approved by the Ethics Committee of Xiangya Hospital, Central South University (Approval No. 2024040420). Written informed consent was obtained from all human participants. The animal study was reviewed and approved by the Animal Ethics Committee of Xiangya Hospital, Central South University (Approval No. 202103138).

## Reporting summary
Further information on research design is available in the Nature Portfolio Reporting Summary linked to this article.

## Data availability
The raw RNA-seq data from mice have been deposited in the Gene Expression Omnibus under accession code GSE295763. The raw RNA-seq data from human NPCs and the raw WGS data from human iPSCs have been deposited in the Genome Sequence Archive for Human under accession code HRA011636 and HRA014016, respectively. The deposition and sharing of the raw data have been approved by the Human Genetics Resource Office in China (registration number: 2025BAT00839). All data needed to evaluate the conclusions in the paper are present in the paper and/or the Supplementary data files. Source data are provided with this paper.

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

## Acknowledgements

The authors would like to thank Peng Jin and Yujing Li from School of Medicine, Emory University; Hao Wu from School of Computer Science and Control Engineering, Shenzhen University of Advanced Technology; Su Yang from Guangdong-Hongkong-Macau Institute of CNS Regeneration, Jinan University; Ying Cheng from Institute of biomedical research, Yunnan University for providing valuable advice to the study. This study was supported by the National Natural Science Foundation of China (82394421 and 82394420 to B.T., 82394422 and 82371874 to X.-J.L., 82271902 and U24A6013 to S.L., 32071037 to Q.L., 82171843 to Y.P., 82101946 to N.X., 82171256 to Q.S.), the National Key R&D Program of China (2021YFA0805200 to H.J.), and the Natural Science Foundation of Hunan Province (2023JJ10097 to Q.L., 2025JJ20079 to Y.P., 2022JJ40832 to N.X.).

## Author contributions

Q.L., Y.P., S.L., and X.-J.L. designed the study, supervised the study, and revised the manuscript; N.X., Y.P., and Q.L. performed experiments, analyzed data, and wrote the manuscript; H.T., Y.L., Y.J., Z.W., J.W., W.Z., X.W., X.S., S.Y., P.Y., Q.S., C.Q., and Y.T. provided important technical assistance to the study; L.S., H.J., D.L., and B.T. provided insightful advice to the study.

## Competing interests

The authors declare no competing interests.
