## [Transparent Peer Review file · Nature Communications]

Precise Excision of Expanded GGC Repeats in NOTCH2NLC via CRISPR/Cas9 for Treating Neuronal Intranuclear Inclusion Disease

Corresponding Author: Professor Qiong Liu

Version 0:

Reviewer comments:

Reviewer #1

(Remarks to the Author)

Summary

This study targets repeat expansion disorders, with a particular focus on neuronal intranuclear inclusion disease (NIID), a recently recognized neurodegenerative disorder. The authors applied Cas9/sgRNA-mediated gene editing to specifically target GGC repeat expansions located in the uORF of NOTCH2NLC, demonstrating therapeutic potential across multiple models by directly addressing the root cause of the disease. The authors rigorously validate their approach across multiple independent platforms, including engineered HEK293 cells, NIID patient-derived induced pluripotent stem cells (iPSCs), and a NIID transgenic mouse model, while incorporating comprehensive molecular, cellular, histological, behavioral, and transcriptomic analyses. Notably, they demonstrate excellent gRNA specificity and off-target control—a critical consideration given the high sequence similarity (>98%) between NOTCH2NLC and other NOTCH2NL family members—supported by robust experimental design and validation techniques such as sequencing and qPCR. In HEK293 cells, they validated the editing efficiency of each sgRNA, confirming that base editing reduces polyG expression, a pathological hallmark of NIID. Next, gene editing and replacement were performed in iPSCs derived from NIID patients, where transcriptomic profiling via volcano plots demonstrated that on-target editing had minimal off-target effects, confirming high specificity. Subsequently, in vivo editing was carried out in a NIID mouse model, with retro-orbital injection at postnatal day 2 followed by behavioral assessments. Edited mice exhibited a reduction in polyG aggregates, alongside decreased GFAP expression and increased NeuN levels, mirroring observations from human cells. Finally, transcriptome-wide changes were analyzed in the mouse model, demonstrating that gene editing rescues transcriptomic dysregulation across both central nervous system and peripheral tissues. Importantly, the study provides compelling evidence for the functional relevance of their intervention by showing clear reversal of disease phenotypes at both the molecular and behavioral levels, with the demonstration of lifespan extension in treated animals marking a major translational milestone. This work not only establishes a foundation for NIID therapy but also holds significant promise for treating other GGC repeat expansion diseases. Although the study is significant and demonstrates important findings, there are comments that authors need to address as listed below.

Abstract

1. Lines 36-38, "Here, we developed a CRISPR/spCas9-based gene-editing strategy to precisely excise the expanded GGC repeats in NOTCH2NL..." To be specific, it should be NOTCH2NLC, not NOTCH2NL.

Introduction

1. Lines 89–97, the description of the function of the NOTCH2NL family genes can be developed more here. Specifically, it is informative to clarify why preserving NOTCH2NL family genes is important—e.g., which one(s) are involved in cortical development and must remain unaltered to avoid developmental deficits."

Results and Figures

1. Line 123, one of the downstream targeted sequences do not match between the main text (864), Fig.1C (864), Fig.S1B (866) and Table.S1 (866). In addition, please describe what the numbers in the names of gRNAs mean. If their names are from their sequence, does the number mean the beginning of the sequences or the end of the sequences or something else. Please also confirm if their names match with their actual number in their sequences. For example, it is puzzling what 702 and 703 are named after since they are not 1 nucleotide apart, which could mis-lead authors intention to readers. Once authors clarify/confirm the nomenclatures, the labels in the cartoon of Figure S1B, main text and the Figure Legends all need to be rechecked and revised accordingly.

2. Lines 126-130, "First, the gene-editing efficiency of each sgRNA was evaluated in human cell lines with normal GGC

repeats (HEK293 cells). Forty-eight hours after co-transfection with Cas9 and sgRNA plasmids (Fig.S1C) at a Cas9-to-sgRNA ratio of 4:1...” what is the rationale behind choosing 4:1? If there is a reference that authors followed, please cite the paper. Otherwise, authors need to briefly clarify why 4:1 was chosen to begin with. Later (Line 145), it is mentioned that “the most effective ratio was found to be 1:1”. Any reason that authors chose to test 1:1, 2:1, 4:1 instead of the opposite ratios for Cas9 to sgRNA (e.g., more sgRNA to Cas9 ratio such as 1:2 or 1:4)? If authors found that 1:1 ratio is most effective among tested ratios, one could think that 1:2 could be even more effective if tested. Please clarify it.

3. Human phenotypes: PolyG aggregation was assessed in both HEK293T cells and mouse tissues, yet it was not evaluated in patient-derived human cells. Given its pathological relevance and the consistency of analysis across models, there appears to be no justification for omitting polyG detection in human fibroblasts. Including this data would strengthen the translational relevance of the findings and enable a more direct comparison across all experimental systems.

4. Figure 2: Are the GFP level alterations demonstrated in this Figure solely due to the GGC shortening or could that be also partially due to the frameshift location after the shortening and leading GFP not be able to be expressed? Please clarify this in the main text. In addition, can the labels of poly G be modified to ‘GFP: PolyG’ instead of ‘PolyG’ as authors did in their previous publication (Sci Adv, 2022) to avoid any misleading/overstatement?

5. Figure 3: The numerical values presented in the text appear consistent overall; however, in Figure 3H and 3I, the sum of individual dots within certain conditions does not exactly match the corresponding group totals. For example, in Figure 3I, the sum of two data points in the “NIID down” group, it is $341 + 903 = 1244$, and in the “GGC-rep down” group, it is $323 + 1394 = 1717$, which aligns well with the indicated totals. However, in other samples or in Figure 3H, the individual data points do not sum precisely to the indicated total. Could the authors clarify whether the plotted values represent raw counts, normalized data, or are derived from averages across replicates? Please revise accordingly and as needed.

6. Figure 3: Can authors indicate what color means what in their Figure legends and can authors consider to use consistent color code for Figure 3I and 3H? For example, you can match the color of NIID down and NIID up between Figure 3I and 3H.

7. Figure 4 and Figure S8: Authors should include age-matched wild-type control (or non-disease control group) to demonstrate the phenotype of NIID mice as their control group (as authors did in Figure 5B-5I). Without comparing with WT mice, authors cannot conclude what authors have demonstrated in treated NIID mice as they rescued the neuropathological phenotypes. Additionally, what are the differences between Figure 4 and Figure S8? Can authors be clear about their description on Figure S8 if there is any difference from Figure 4? Even if authors wanted to demonstrate Figure S8 as another representative Figure to support Figure 4, it is unclear if the Figure S8 was representative images as a part of the quantification result presented in Figure 4 or if Figure S8 was an independent experiment. If Figure S8 was an independent experiment, please have their quantified result as a graph to support author’s conclusion about Figure S8. If not, please clarify in the main text and in the Figure legends.

8. Figure 5J: The correlation between zsGreen expression, polyG reduction, and GFAP/NeuN changes appears inconsistent, making it difficult to link editing efficiency with phenotypic rescue. For example, second sample of Treated NIID mice has higher expression levels than other two mice, but their GFAP or NeuN levels looks fairly similar to each other between all 3 mice compared to how different they are from the second mouse. Authors need to clarify how or whether they normalized their data to the levels of zsGreen. Comparing samples between mice which demonstrate comparable zsGreen expression levels would improve data interpretation and strengthen conclusions.

9. Figure 5K lacks a non-disease control group, making it difficult to evaluate the full extent of pathological elevation or rescue. The authors should include age-matched wild-type control group to establish a baseline and demonstrate whether treatment restores polyG to the physiological levels.

10. Figure S1A: The second ‘Upstream mismatch sequences’ should be revised to be ‘Downstream mismatch sequences’.

11. Figure S3B: The X-axis labels indicate Untreated vs Treated. However, the description is missing in the Figure and Figure Legends about what they were treated with. More detail information about Figure S3B needs to be added to the Figure Legends. In addition, the X-axis labels need to be revised to make it clearer and consistent to other labels.

12. Figure S4: The order in the main text and the Figure is RP-PCR then GC-PCR, which doesn’t match with the Figure Legend (GC-PCR then RP-PCR). Please revise the Figure Legend accordingly to match with other part of the manuscript.

13. Figure S10A: Y-axis label is missing from the graph.

Methods

1. For animal study, were both sexes used, or only male or only female? If so, can authors describe it clearly in the method section and justify the reason why they chose those genders.
2. Antibody information: It is better to indicate dilution ratios for primary and secondary antibodies to increase reproducibility of the study.
3. Were any mortality or adverse events occurred during stereotaxic surgery or during the behavior with animals who went through surgeries? If so, please describe it in the manuscript.
4. Information of patient age/sex and number of clones analyzed would strengthen rigor. Please describe them as much as authors can.

Reviewer #2

(Remarks to the Author)

In this work, Xie et al. have used a CRISPR/Cas9-based gene-editing approach to remove the expanded GGC repeats in the NOTCH2NLC gene, which is the known genetic cause of neuronal intranuclear inclusion disease (NIID). To prevent off-target effects in homologous genes, the authors created and verified dual sgRNAs that flank the repeat region. HEK293 cells, NIID patient-derived iPSCs, and a transgenic mouse model of NIID were used to validate the editing technique. The findings demonstrated that the expansion of the repeat was efficiently removed, that pathological polyG aggregates were reduced, that transcriptomic profiles were restored, and that the neurobehavioral phenotypes in mice were reversed. The work has great clinical translation potential and is backed by lots of phenotypic and molecular data.

The manuscript is strong in several ways. It presents a very specific, cleverly designed gene-editing technique and has been

proven to work in a variety of biological systems. The findings' robustness and translational potential are enhanced by the use of in vitro, ex vivo, and in vivo models. The paper offers a comprehensive examination that includes behavioral evaluations in animal models, transcriptome rescue, and off-target consequences. Furthermore, the work addresses important translational challenges like immune response and vector administration within a therapeutically relevant framework.

Overall the study is well planned and carried out. The main conclusions are sound, novel, and I think that this work is of high interest to the fields of genome editing and neurogenetics. I only have a few suggestions/remarks. First, only the top 14 projected loci are included in the off-target prediction. This could be improved by more thorough genome-wide off-target studies, such as GUIDE-seq or CIRCLE-seq. Maybe the authors could discuss this. Second, although the mouse model exhibits important characteristics, its limitations in replicating the human disease should be discussed in more detail. Third, it would be good to include a summary table for sgRNAs, editing efficiencies, and off-target profiles.

In summary, my recommendation is accept with minor revisions.

Reviewer #3

(Remarks to the Author)

Version 1:

Reviewer comments:

Reviewer #1

(Remarks to the Author)

Authors addressed all of my concerns. The readability and the solidity of the manuscript improved significantly.

Reviewer #2

(Remarks to the Author)

The authors have thoroughly addressed my comments and have significantly improved the manuscript through their revisions. I have no further suggestions and recommend the manuscript for publication.

Reviewer #3

(Remarks to the Author)

Responses to reviewers' comments

We sincerely thank the reviewers for their highly positive comments, their recognition of the significance and insightfulness of our work. Their constructive critiques have been invaluable in refining our manuscript. We have thoroughly addressed all concerns raised, revising the text point-by-point and incorporating corresponding revisions throughout.

In addition, we have integrated the new perspectives suggested by the reviewers and refined the Discussion section to enhance its clarity and to more effectively convey the significance of our findings. We have highlighted all changes in blue in the revised manuscript, and believe these revisions have significantly improved the overall quality of the manuscript. We hope the reviewers and editor now consider it suitable for publication in *Nature Communications*.

The detailed responses are shown below (in blue font).

Reviewer comments

Reviewer #1 (Remarks to the Author):

Summary

This study targets repeat expansion disorders, with a particular focus on neuronal intranuclear inclusion disease (NIID), a recently recognized neurodegenerative disorder. The authors applied Cas9/sgRNA-mediated gene editing to specifically target GGC repeat expansions located in the uORF of NOTCH2NLC, demonstrating therapeutic potential across multiple models by directly addressing the root cause of the disease. The authors rigorously validate their approach across multiple independent platforms, including engineered HEK293 cells, NIID patient-derived induced pluripotent stem cells (iPSCs), and a NIID transgenic mouse model, while incorporating comprehensive molecular, cellular, histological, behavioral, and transcriptomic analyses. Notably, they demonstrate excellent gRNA specificity and off-target control—a critical consideration given the high sequence similarity (>98%) between NOTCH2NLC and other NOTCH2NL family members—supported by robust experimental design and validation techniques such as sequencing and qPCR. In HEK293 cells, they validated

the editing efficiency of each sgRNA, confirming that base editing reduces polyG expression, a pathological hallmark of NIID. Next, gene editing and replacement were performed in iPSCs derived from NIID patients, where transcriptomic profiling via volcano plots demonstrated that on-target editing had minimal off-target effects, confirming high specificity. Subsequently, in vivo editing was carried out in a NIID mouse model, with retro-orbital injection at postnatal day 2 followed by behavioral assessments. Edited mice exhibited a reduction in polyG aggregates, alongside decreased GFAP expression and increased NeuN levels, mirroring observations from human cells. Finally, transcriptome-wide changes were analyzed in the mouse model, demonstrating that gene editing rescues transcriptomic dysregulation across both central nervous system and peripheral tissues. Importantly, the study provides compelling evidence for the functional relevance of their intervention by showing clear reversal of disease phenotypes at both the molecular and behavioral levels, with the demonstration of lifespan extension in treated animals marking a major translational milestone. This work not only establishes a foundation for NIID therapy but also holds significant promise for treating other GGC repeat expansion diseases. Although the study is significant and demonstrates important findings, there are comments that authors need to address as listed below.

Author response: Thank you so much for the time and efforts spent on reviewing our manuscript. We sincerely appreciate your positive and encouraging comments.

Abstract

1. Lines 36-38, "Here, we developed a CRISPR/spCas9-based gene-editing strategy to precisely excise the expanded GGC repeats in NOTCH2NL..." To be specific, it should be NOTCH2NLC, not NOTCH2NL.

Author response: Thank you for pointing this out. The correct gene name should be *NOTCH2NLC*, and we have made the corresponding revision in the text.

Introduction

1. Lines 89–97, the description of the function of the NOTCH2NL family genes can be developed more here. Specifically, it is informative to clarify why preserving NOTCH2NL family genes is important—e.g., which one(s) are involved in cortical development and must remain unaltered to avoid developmental deficits.”

Author response: Thank you for the suggestions. The *NOTCH2NL* gene family plays critical roles in human brain development and homeostasis. Specifically, *NOTCH2NLA*, *NOTCH2NLB*, and *NOTCH2NLC* are expressed in cortical progenitors, where they enhance self-renewal and delay premature differentiation by potentiating canonical Notch signaling, thereby contributing to the evolutionary expansion of the human cerebral cortex. Abnormal dosage of *NOTCH2NL* genes is associated with the 1q21.1 deletion/duplication syndrome. Deletions are linked to microcephaly or cortical malformations, while duplications correlate with macrocephaly or autism spectrum disorders¹⁻³. Thus, it is important to preserve the functions of *NOTCH2NL* family genes.

We have incorporated this information in the revised text as follows:

The human-specific *NOTCH2NL* (*NOTCH2* N-terminal like) gene family, which includes *NOTCH2NLA*, *NOTCH2NLB*, *NOTCH2NLC* and *NOTCH2NLR*, shares high sequence homology and plays critical roles in human brain development and homeostasis³⁻⁵. Specifically, *NOTCH2NLA*, *NOTCH2NLB*, and *NOTCH2NLC* are expressed in cortical progenitors, where they enhance self-renewal and delay premature differentiation by potentiating canonical Notch signaling, thereby contributing to the evolutionary expansion of the human cerebral cortex. Abnormal dosage of *NOTCH2NL* genes has been linked to neurodevelopmental disorders, including microcephaly or cortical malformations upon loss and macrocephaly or autism spectrum disorders upon duplication. Therefore, it is essential to specifically target the GGC repeat expansion within *NOTCH2NLC* while preserving the normal expression and function of the *NOTCH2NL* family. However, the near-identical sequence flanking the GGC repeats (<2% divergence) among these homologous genes, combined with the large size and high GC content of the expanded repeats, pose significant challenges for precise and efficient gene editing.

Results and Figures

1. Line 123, one of the downstream targeted sequences do not match between the main text (864), Fig.1C (864), Fig.S1B (866) and Table.S1 (866). In addition, please describe what the numbers in the names of gRNAs mean. If their names are from their sequence, does the number mean the beginning of the sequences or the end of the sequences or something else. Please also confirm if their names match with their actual number in

their sequences. For example, it is puzzling what 702 and 703 are named after since they are not 1 nucleotide apart, which could mis-lead authors intention to readers. Once authors clarify/confirm the nomenclatures, the labels in the cartoon of Figure S1B, main text and the Figure Legends all need to be rechecked and revised accordingly.

Author response: Thank you for your careful scrutiny of our manuscript. The designation of sgRNA864 was a typographical error and should have been sgRNA866. We apologize for the confusion caused by this misleading nomenclature. The original names of the sgRNAs were based on their physical cleavage positions on chromosome 1 of the GRCh38/hg38 reference genome. To avoid ambiguity, we have now renamed them sequentially as sgRNA1-8. In addition, we provide a new table summarizing detailed information for each sgRNA, including genomic coordinates, strand orientation, and other relevant features. We have carefully rechecked and revised the labels in the main text, figure legends, and figures according to the corrected nomenclature.

The new Supplementary Table 1 is shown below:

Name	Cut position	Strand	Sequence	PAM	On-target efficiency (%)	Off-target profile
sgRNA1	149390717	+	TGCTTCGGACCGTAGCG CCA	GGG	76	NOTCH2NLC -specific
sgRNA2	149390716	+	GTGCTTCGGACCGTAGC GCC	AGG	46	Not evaluated
sgRNA3	149390702	-	GGCGCTACGGTCCGAAG CAC	AGG	63	Not evaluated
sgRNA4	149390702	+	AGGCATTTGCGCCTGTG CTT	CGG	60	Not evaluated
sgRNA5	149390874	+	CGCCCTGCGCCGCTCTGC TG	TGG	56	NOTCH2NLC -specific
sgRNA6	149390865	-	GCCCACAGCAGAGCGGC GCA	GGG	57	NOTCH2NLC -specific
sgRNA7	149390866	-	CGCCCACAGCAGAGCGG CGC	AGG	43	Not evaluated
sgRNA8	149390862	-	CACAGCAGAGCGGCGCA GGG	CGG	39	Not evaluated

2. Lines 126-130, “First, the gene-editing efficiency of each sgRNA was evaluated in

human cell lines with normal GGC repeats (HEK293 cells). Forty-eight hours after co-transfection with Cas9 and sgRNA plasmids (Fig.S1C) at a Cas9-to-sgRNA ratio of 4:1...” what is the rationale behind choosing 4:1? If there is a reference that authors followed, please cite the paper. Otherwise, authors need to briefly clarify why 4:1 was chosen to begin with.

Author response: The 4:1 Cas9-to-sgRNA ratio was selected for preliminary screen based on our laboratory's prior experience with genome editing in repeat expansion disorders. We have now cited a relevant reference supporting this choice⁶, as it represents a standard practice in our lab for preliminary efficiency testing of different sgRNAs.

Later (Line 145), it is mentioned that “the most effective ratio was found to be 1:1”. Any reason that authors chose to test 1:1, 2:1, 4:1 instead of the opposite ratios for Cas9 to sgRNA (e.g., more sgRNA to Cas9 ratio such as 1:2 or 1:4)? If authors found that 1:1 ratio is most effective among tested ratios, one could think that 1:2 could be even more effective if tested. Please clarify it.

Author response: We appreciate your insightful comment. After identifying the effective individual sgRNA, we subsequently examined different combinations at varying ratios. Our experimental results (Supplementary Fig. 2C) demonstrated that, among the three ratios we tested (4:1, 2:1, and 1:1), the 1:1 ratio was the most effective when targeting both upstream and downstream sites simultaneously. We did not test lower Cas9 to sgRNA ratio (such as 1:2 or 1:4) for the following reasons. First, the coding sequence of Cas9 (~4.2kb) is much larger than that of sgRNA (<0.1kb). Therefore, under equivalent transfection conditions, a given mass of plasmid DNA may yield a substantially higher molar amount of sgRNA than Cas9. Second, previous studies have suggested that an excess of sgRNA may increase off-target effects⁷⁻⁹.

3. Human phenotypes: PolyG aggregation was assessed in both HEK293T cells and mouse tissues, yet it was not evaluated in patient-derived human cells. Given its pathological relevance and the consistency of analysis across models, there appears to be no justification for omitting polyG detection in human fibroblasts. Including this data would strengthen the translational relevance of the findings and enable a more direct comparison across all experimental systems.

Author response: Thank you for the invaluable suggestion. To date, it remains unclear whether there are intranuclear inclusions in cultured skin fibroblasts of NIID patient. To follow this advice, we examined polyG inclusions in cultured fibroblasts derived from a NIID patient (carrying 97/22 GGC repeats in *NOTCH2NLC*) and a health control (with 16/20 GGC repeats). Two antibodies were employed: NOTCH2NLC-polyG (a specific antibody verified in our previous study¹⁰) and P62 (widely used in clinical pathological diagnosis) were used. We did not observe the specific presence of polyG puncta in the NIID patient-derived fibroblasts. It is possible that the polyG inclusions are absent in the cultured skin fibroblasts under the *in vitro* condition. Alternatively, the available antibody we used may not specifically or sensitively label polyG aggregates in human fibroblasts due to cell-type specific modifications. Because of these possible reasons, we were unable to confidently confirm the presence of polyG inclusions in fibroblasts and did not proceed with testing polyG inclusions in cultured human fibroblasts. Please see the following figures that are not included in the revision.

[Figure Redacted]

4. Figure 2: Are the GFP level alterations demonstrated in this Figure solely due to the GGC shortening or could that be also partially due to the frameshift location after the shortening and leading GFP not be able to be expressed? Please clarify this in the main text. In addition, can the labels of poly G be modified to 'GFP: PolyG' instead of 'PolyG' as authors did in their previous publication (Sci Adv, 2022) to avoid any misleading/overstatement?

Author response: We appreciate your thoughtful comment. We agree that the lack of GFP expression in the stable cells could result either from GGC repeat deletion or from disruption by an out-frame shift. To distinguish these possibilities, we designed a pair of primers: the forward primer was located upstream of the upstream targeting region, and the reverse primer was within the GFP sequence. This primer pair specifically amplifies the exogenous *NOTCH2NLC*-98GGC-GFP fragment without amplifying the endogenous *NOTCH2NLC* gene.

Agarose gel electrophoresis showed that PCR amplification from the genomic DNA of the 98polyG stable cell line treated with Cas9 and control sgRNA yielded a single 587 bp band, corresponding to the allele carrying 98 GGC repeats. In contrast, cells treated with Cas9 and sgRNA1+6 exhibited additional 187 bp and 350 bp bands (Supplementary Fig. 3A). Subsequent TA cloning and Sanger sequencing revealed that the 187 bp product resulted from complete excision of the sequence between the two sgRNA target sites, whereas the 350 bp product contained approximately 22 GGC repeats (within the normal range) and maintained the in-frame GFP fusion (Supplementary Fig. 3B-C). Sequencing of the 587 bp band confirmed that none of the 15 analyzed TA clones exhibited a frameshift mutation (Supplementary Fig. 3D). Collectively, these findings indicate that the reduction of polyG-GFP protein levels in the stable cell line primarily results from deletion or contraction of the expanded GGC repeats rather than frameshift mutations. The GGC deletion efficiency was approximately 60% (Supplementary Fig. 3E).

Additionally, we updated the label of polyG to “GFP: PolyG” in Figure 2 to maintain consistency with our previous publication and avoid potential misinterpretation or overstatement.

Supplementary Figure 4

5. Figure 3: The numerical values presented in the text appear consistent overall; however, in Figure 3H and 3I, the sum of individual dots within certain conditions does not exactly match the corresponding group totals. For example, in Figure 3I, the sum of two data points in the ‘‘NIID down’’ group, it is $341 + 903 = 1244$, and in the ‘‘GGC-rep down’’ group, it is $323 + 1394 = 1717$, which aligns well with the indicated totals. However, in other samples or in Figure 3H, the individual data points do not sum precisely to the indicated total. Could the authors clarify whether the plotted values represent raw counts, normalized data, or are derived from averages across replicates? Please revise accordingly and as needed.

Author response: Thank you for carefully reviewing our data presentation. Upon re-examination, we confirm that the values plotted in these Upset plot figures represent the number of differentially expressed genes (DEGs) identified from the comparison groups shown in Figure 3E-G. These values are not raw sequencing counts (e.g., raw reads, FPKM, or TPM), but rather the results of bioinformatic analysis based on three biological replicates per group and defined using the thresholds: $|\log_2(\text{fold change})| \geq 2$ and $\text{FDR} < 0.05$.

The discrepancies between the sum of individual data and the indicated group totals were due to incomplete data display. In original Figure 3H, 16 genes upregulated both in the GGC-deletion-NPC and NIID NPC, and 29 genes downregulated both in the GGC-deletion-NPC and NIID NPC were not shown. In original Figure 3I, 18 genes upregulated both in the GGC-repair-NPC and NIID NPC were not shown. We have supplemented the omitted data in the revised figures. The individual data now accurately sum up to the indicated group totals, ensuring the consistency and accuracy of the data presentation. The updated Figure 3H and 3I are shown below.

Updated Figure 5H-I

In addition, in original Supplementary Figure 10G, one gene consistently upregulated and three genes consistently downregulated in both the treat and untreated group were not shown. We have supplemented the omitted data in the revised figure. The individual data now accurately sum up to the indicated group totals. The updated Supplementary Figure 12G is shown below.

Updated Supplementary Figure 12G

6. Figure 3: Can authors indicate what color means what in their Figure legends and can authors consider to use consistent color code for Figure 3I and 3H? For example, you can match the color of NIID down and NIID up between Figure 3I and 3H.

Author response: Thank you for the invaluable suggestion. We have updated the figure legends to clearly indicate what each color represents. The dark blue and dark red rectangles indicated genes down-regulated and up-regulated in the unedited NIID NPC, respectively. The light blue and pink rectangles indicated genes down-regulated and up-regulated in the edited NIID NPC, respectively. Additionally, we have made the color code consistent for Figure 3I and 3H, specifically matching the colors for NIID down and NIID up in these two figures.

7. Figure 4 and Figure S8: Authors should include age-matched wild-type control (or non-disease control group) to demonstrate the phenotype of NIID mice as their control group (as authors did in Figure 5B-5I). Without comparing with WT mice, authors cannot conclude what authors have demonstrated in treated NIID mice as they rescued the neuropathological phenotypes.

Additionally, what are the differences between Figure 4 and Figure S8? Can authors be clear about their description on Figure S8 if there is any difference from Figure 4? Even if authors wanted to demonstrate Figure S8 as another representative Figure to support Figure 4, it is unclear if the Figure S8 was representative images as a part of the quantification result presented in Figure 4 or if Figure S8 was an independent experiment. If Figure S8 was an independent experiment, please have their quantified result as a graph to support author's conclusion about Figure S8. If not, please clarify in the main text and in the Figure legends.

Author response: Thank you for your thoughtful comments. The original Figure S8 consists of the same cohort of mice as Figure 4, but shown at lower magnifications. Since these images did not provide additional information, we have removed it in the revision.

The experiment in Figure 4 was designed as an initial *in vivo* test of whether localized knockout of the GGC repeat could ameliorate the pathological phenotype, before proceeding with retro-orbital venous injection. To minimize inter-individual

variability, therapeutic and control AAV were injected into the left and right striatum of the same NIID mouse, respectively. As you pointed out, this experiment alone does not allow us to fully quantify the extent of recovery.

Following your suggestion, we have now included a new Supplementary Figure 10 with age-matched non-disease control group. This new figure clearly demonstrates the pathological hallmarks of NIID mice and shows that gene therapy markedly ameliorated these abnormalities. Specifically, polyG inclusions were almost completely cleared to levels comparable to the non-disease controls, whereas NeuN expression and gliosis exhibited substantial but partial recovery.

New Supplementary Figure 10

8. Figure 5J: The correlation between zsGreen expression, polyG reduction, and GFAP/NeuN changes appears inconsistent, making it difficult to link editing efficiency with phenotypic rescue. For example, second sample of Treated NIID mice has higher expression levels than other two mice, but their GFAP or NeuN levels looks fairly similar to each other between all 3 mice compared to how different they are from the second mouse. Authors need to clarify how or whether they normalized their data to the

levels of zsGreen. Comparing samples between mice which demonstrate comparable zsGreen expression levels would improve data interpretation and strengthen conclusions.

Author response: Thank you for the invaluable suggestion. In the original Figure 5J, polyG, GFAP and NeuN protein levels were normalized to vinculin rather than zsGreen. We agree that comparing mice with comparable zsGreen expression levels would improve the interpretation. Accordingly, we reanalyzed data by including additional mice with comparable zsGreen expression levels. The updated Western blot results demonstrated that, compared to the untreated NIID mice, treated NIID mice showed significantly reduced protein levels of polyG and GFAP, and increased NeuN levels in the cortex. Notably, GFAP and NeuN expression in treated NIID mice were largely restored to the levels in control mice (new Fig.5J-K). We acknowledge that some variability in zsGreen expression was still observed across groups, which may reflect differences in AAV delivery efficiency among individual mice. Nonetheless, the consistent directionality of changes in polyG, GFAP, and NeuN strongly supports our conclusion that the CRISPR/Cas9 editing system effectively mitigates neurodegenerative phenotypes in NIID mice.

The updated results are shown below (The treated group is NIID mice injected with AAV-Cas9 and AAV-sgRNA1+6; the untreated group is NIID mice injected with AAV-Cas9 and AAV-control sgRNA; the control group is littermate control mice injected with AAV-Cas9 and AAV-sgRNA).

Updated Figure 5J-K

9. Figure 5K lacks a non-disease control group, making it difficult to evaluate the full extent of pathological elevation or rescue. The authors should include age-matched wild-type control group to establish a baseline and demonstrate whether treatment restores polyG to the physiological levels.

Author response: Thank you for this insightful suggestion. We have now included an age-matched non-disease control group in the revised Figure 5J-K (see figure above). By adding this group, we were able to establish a clear baseline for the normal levels of polyG, GFAP, and NeuN. Our results showed that therapeutic AAV treatment in NIID mice restored polyG aggregation, GFAP and NeuN to the levels in non-disease control mice.

10. Figure S1A: The second ‘Upstream mismatch sequences’ should be revised to be ‘Downstream mismatch sequences’.

Author response: Thank you for your careful scrutiny of our manuscript. We have revised the second “Upstream mismatch sequences” in Figure S1A to “Downstream mismatch sequences” as suggested.

11. Figure S3B: The X-axis labels indicate Untreated vs Treated. However, the description is missing in the Figure and Figure Legends about what they were treated with. More detail information about Figure S3B needs to be added to the Figure Legends. In addition, the X-axis labels need to be revised to make it clearer and consistent to other labels.

Author response: Thank you for your careful scrutiny of our manuscript. We have added detailed information about what the samples were treated with in the Figure Legends of Figure S3B. Also, we have revised the X-axis labels to make them clearer and consistent with other labels.

12. Figure S4: The order in the main text and the Figure is RP-PCR then GC-PCR, which doesn’t match with the Figure Legend (GC-PCR then RP-PCR). Please revise the Figure Legend accordingly to match with other part of the manuscript.

Author response: Thank you for your careful scrutiny of our manuscript. We have revised the Figure Legend of Figure S4 to match the order of RP-PCR then GC-PCR, which is consistent with the main text and the Figure.

13. Figure S10A: Y-axis label is missing from the graph.

Author response: Thank you for your careful scrutiny of our manuscript. We have added the Y-axis label to the graph in Figure S10A.

Methods

1. For animal study, were both sexes used, or only male or only female? If so, can authors describe it clearly in the method section and justify the reason why they chose those genders.

Author response: In this study, both male and female mice were used in all experiments. We did not observe sex-dependent differences in our analyses, which is consistent with previous reports showing no significant differences in pathological or behavioral phenotypes between males and females¹⁰.

2. Antibody information: It is better to indicate dilution ratios for primary and secondary antibodies to increase reproducibility of the study.

Author response: We have now added dilution ratios in the method section for all primary and secondary antibodies in the Methods section to improve the reproducibility of our study.

3. Were any mortality or adverse events occurred during stereotaxic surgery or during the behavior with animals who went through surgeries? If so, please describe it in the manuscript.

Author response: During stereotaxic surgery, no significant adverse events were observed in P8-P10 mice receiving intrastriatal stereotaxic injections. All stereotaxic surgeries were completed successfully, and surgical incisions healed satisfactorily without adverse events. The injected mice survived and underwent subsequent

pathological analyses without complications.

For retro-orbital venous injection, sixty-four neonatal P1-P2 pups were injected with AAV and less than ten pups died, likely due to poor rewarming after hypothermic anesthesia or cannibalized by the dam. The remaining pups survived and successfully underwent subsequent experimental procedures.

We have included this information in Methods section.

4. Information of patient age/sex and number of clones analyzed would strengthen rigor. Please describe them as much as authors can.

Author response: Thank you for the suggestion. The patient was a 57-year-old male who has presented with bradykinesia and tremors in bilateral upper extremities for over ten years. Physical examination revealed increased muscle tone in all four extremities and action tremors in bilateral upper extremities. Cognitive assessment yielded an MMSE score of 28 and a MoCA score of 27. The patient declined skin biopsy. Genetic testing indicated a *NOTCH2NLC* gene carrying 113 and 17 GGC repeats. In addition, we analyzed ninety iPSC clones in total, with four clones with the GGC repeats deleted or repaired.

We have added this information in the revised main text.

Reviewer #2 (Remarks to the Author):

In this work, Xie et al. have used a CRISPR/Cas9-based gene-editing approach to remove the expanded GGC repeats in the *NOTCH2NLC* gene, which is the known genetic cause of neuronal intranuclear inclusion disease (NIID). To prevent off-target effects in homologous genes, the authors created and verified dual sgRNAs that flank the repeat region. HEK293 cells, NIID patient-derived iPSCs, and a transgenic mouse model of NIID were used to validate the editing technique. The findings demonstrated that the expansion of the repeat was efficiently removed, that pathological polyG aggregates were reduced, that transcriptomic profiles were restored, and that the neurobehavioral phenotypes in mice were reversed. The work has great clinical translation potential and is backed by lots of phenotypic and molecular data. The manuscript is strong in several ways. It presents a very specific, cleverly designed gene-

editing technique and has been proven to work in a variety of biological systems. The findings' robustness and translational potential are enhanced by the use of *in vitro*, *ex vivo*, and *in vivo* models. The paper offers a comprehensive examination that includes behavioral evaluations in animal models, transcriptome rescue, and off-target consequences. Furthermore, the work addresses important translational challenges like immune response and vector administration within a therapeutically relevant framework. Overall the study is well planned and carried out. The main conclusions are sound, novel, and I think that this work is of high interest to the fields of genome editing and neurogenetics.

Author response: Thank you very much for your thoughtful and encouraging comments. We greatly appreciate your recognition of the novelty and translational relevance of our study, which motivates us to further pursue this line of research.

I only have a few suggestions/remarks. First, only the top 14 projected loci are included in the off-target prediction. This could be improved by more thorough genome-wide off-target studies, such as GUIDE-seq or CIRCLE-seq. Maybe the authors could discuss this.

Author response: We appreciate your insightful suggestion regarding off-target analysis and fully agree that unbiased genome-wide approaches are critical for assessing genome-editing safety. CIRCLE-seq and GUIDE-seq are two widely adopted methods with distinct advantages. CIRCLE-seq offers ultra-sensitive *in vitro* detection of potential off-target sites and is particularly valuable for sgRNA pre-screening¹¹. However, because it is performed on naked DNA, it may overestimate off-target activity and does not reflect chromatin context. GUIDE-seq, by contrast, captures off-target events in the native cellular environment, providing high sensitivity and clinical relevance¹². Nevertheless, it requires specialized expertise and optimized protocols that were not available to us in the present study. For these reasons, we did not apply CIRCLE-seq or GUIDE-seq here, but we acknowledge their importance and plan to incorporate these unbiased methods in future work.

To improve off-target assays, we combined targeted deep-seq and WGS to evaluate potential off-target activity in the revision. For deep-seq, we increased biological replicates and extended coverage to the top 20 predicted sites, and consistently

confirmed that editing was restricted to *NOTCH2NLC* (revised Fig. 3A and new Supplementary Fig. 8A-B).

New Supplementary Figure 8

Furthermore, we performed 50× WGS to assess overall genomic integrity and potential off-target mutations. WGS is considered a comprehensive approach for identifying SNVs, indels, CNVs, and structural variants¹³. Our results revealed comparable and negligible extent of single-nucleotide variants (SNVs), structure variants (SVs) and indels between edited iPSCs and unedited iPSC (new Supplementary Fig. 8C). In addition, read-depth analysis of the additional 30 predicted off-target sites also showed no significant differences between groups (new Supplementary Fig.8D-E), further supporting the specificity of our editing strategy.

While the sensitivity of WGS for detecting extremely rare off-target events is generally lower than that of CIRCLE-seq or GUIDE-seq, the combination of targeted deep-seq and WGS offered complementary advantages, enabling a comprehensive assessment of editing specificity in this study. Incorporating unbiased, genome-wide approaches such as GUIDE-seq in future work will be essential for comprehensive off-target assessment prior to clinical translation.

Second, although the mouse model exhibits important characteristics, its limitations in replicating the human disease should be discussed in more detail.

Author response: Thank you for your invaluable comment. The transgenic NIID mouse model used in this study recapitulates several key pathological features of NIID, including polyG aggregates, neuroinflammation, and motor deficits¹⁰, making it an important tool for evaluating therapeutic efficacy. However, this model also has some limitations.

One major limitation is its rapid disease progression: pathological and behavioral abnormalities emerge around 40 days of age, and the animals typically die prematurely by around two months¹⁰. This is in sharp contrast to human patients, who usually develop symptoms between 30 and 60 years of age and may survive for decades after onset¹⁴. This discrepancy may stem from the fact that the *NOTCH2NLC* gene is human-specific, and its expression in rodent models may exacerbate the toxicity of expanded GGC repeats and accelerate disease progression due to the absence of native regulatory pathways in mice.

Another limitation is that the current mouse model fails to reproduce key magnetic resonance imaging (MRI) features characteristic of NIID, such as hyperintensity signals along the corticomedullary junction on diffusion-weighted imaging^{14,15}. This difference may be attributed to the simpler white matter architecture and lower glial cell density in mouse brains compared with humans. In contrast, large-animal models, particularly non-human primates, may better recapitulate these MRI feature owing to their closer genetic, physiological, and neuroanatomical similarities to humans, as well as their higher proportion of glial cells¹⁶. Therefore, the development of large-animal NIID models, in combination with advanced neuroimaging approaches, represents a promising strategy for more faithfully modeling human disease and accelerating translational research, particularly for evaluating the efficacy, safety, and delivery efficiency of gene-editing therapies in a system that more closely resembles the human brain.

We have incorporated this information in revised Discussion.

Third, it would be good to include a summary table for sgRNAs, editing efficiencies, and off-target profiles.

Author response: We have added a summary table including sgRNAs sequences, cutting position, editing efficiencies, and off-target profiles in the revised supplemental files. In addition, we have renamed the sgRNAs to avoid ambiguity.

The new Supplementary Table 1 was shown as below:

Name	Cut position	Strand	Sequence	PAM	On-target efficiency (%)	Off-target profile
sgRNA1	149390717	+	TGCTTCGGACCGTAGCG CCA	GGG	76	NOTCH2NLC -specific
sgRNA2	149390716	+	GTGCTTCGGACCGTAGC GCC	AGG	46	Not evaluated
sgRNA3	149390702	-	GGCGCTACGGTCCGAAG CAC	AGG	63	Not evaluated
sgRNA4	149390702	+	AGGCATTTGCGCCTGTG CTT	CGG	60	Not evaluated
sgRNA5	149390874	+	CGCCCTGCGCCGCTCTGC TG	TGG	56	NOTCH2NLC -specific

sgRNA6	149390865	-	GCCCACAGCAGAGCGGC GCA	GGG	57	NOTCH2NLC -specific
sgRNA7	149390866	-	CGCCCACAGCAGAGCGG CGC	AGG	43	Not evaluated
sgRNA8	149390862	-	CACAGCAGAGCGGCGCA GGG	CGG	39	Not evaluated

In summary, my recommendation is accept with minor revisions.

Author response: Thank you very much for your positive evaluation and recommendation. We sincerely appreciate your encouraging comments.

Reviewer #3 (Remarks to the Author):

Author response: We sincerely appreciate your positive comments, careful review, and valuable suggestions, which have been invaluable in refining our manuscript.

Reference

1. Fiddes, I.T. *et al.* Human-Specific NOTCH2NL Genes Affect Notch Signaling and Cortical Neurogenesis. *Cell* **173**, 1356-1369.e22 (2018).
2. Suzuki, I.K. *et al.* Human-Specific NOTCH2NL Genes Expand Cortical Neurogenesis through Delta/Notch Regulation. *Cell* **173**, 1370-1384.e16 (2018).
3. Florio, M. *et al.* Evolution and cell-type specificity of human-specific genes preferentially expressed in progenitors of fetal neocortex. *eLife* **7**(2018).
4. Fiddes, I.T. *et al.* Human-Specific NOTCH2NL Genes Affect Notch Signaling and Cortical Neurogenesis. *Cell* **173**, 1356-+ (2018).
5. Suzuki, I.K. *et al.* Human-Specific NOTCH2NL Genes Expand Cortical Neurogenesis through Delta/Notch Regulation. *Cell* **173**, 1370-+ (2018).

6. Yang, S. *et al.* CRISPR/Cas9-mediated gene editing ameliorates neurotoxicity in mouse model of Huntington's disease. *J Clin Invest* **127**, 2719-2724 (2017).
7. Kim, S., Kim, D., Cho, S.W., Kim, J. & Kim, J.-S. Highly efficient RNA-guided genome editing in human cells via delivery of purified Cas9 ribonucleoproteins. *Genome Research* **24**, 1012-1019 (2014).
8. Fu, Y. *et al.* High-frequency off-target mutagenesis induced by CRISPR-Cas nucleases in human cells. *Nature Biotechnology* **31**, 822-826 (2013).
9. Hsu, P.D. *et al.* DNA targeting specificity of RNA-guided Cas9 nucleases. *Nature Biotechnology* **31**, 827-832 (2013).
10. Liu, Q. *et al.* Expression of expanded GGC repeats within NOTCH2NLC causes behavioral deficits and neurodegeneration in a mouse model of neuronal intranuclear inclusion disease. *Sci Adv* **8**, eadd6391 (2022).
11. Tsai, S.Q. *et al.* CIRCLE-seq: a highly sensitive in vitro screen for genome-wide CRISPR–Cas9 nuclease off-targets. *Nature Methods* **14**, 607-614 (2017).
12. Tsai, S.Q. *et al.* GUIDE-seq enables genome-wide profiling of off-target cleavage by CRISPR-Cas nucleases. *Nature Biotechnology* **33**, 187-197 (2015).
13. Anderson, K.R. *et al.* CRISPR off-target analysis in genetically engineered rats and mice. *Nature Methods* **15**, 512-514 (2018).
14. Tian, Y. *et al.* Clinical features of NOTCH2NLC-related neuronal intranuclear inclusion disease. *J Neurol Neurosurg Psychiatry* **93**, 1289-1298 (2022).
15. Tai, H. *et al.* Clinical Features and Classification of Neuronal Intranuclear Inclusion Disease. *Neurol Genet* **9**, e200057 (2023).
16. Yin, P., Li, S., Li, X.J. & Yang, W. New pathogenic insights from large animal models of neurodegenerative diseases. *Protein Cell* **13**, 707-720 (2022).

Responses to referees' and editors' comments

We sincerely thank the reviewers for their time, insightful comments, and constructive suggestions, which have significantly improved the quality of our work.

In this version, we have thoroughly reviewed the manuscript and supplementary materials against the journal's checklist and have implemented all necessary revisions to ensure full compliance (all changes are highlighted in blue in the manuscript). We believe the manuscript now fully meets the journal's standards and hope it is deemed suitable for publication.

The detailed responses are shown below.

Reviewer comments

Reviewer #1 (Remarks to the Author):

Authors addressed all of my concerns. The readability and the solidity of the manuscript improved significantly.

We thank the reviewer for their positive feedback and for confirming that our revisions have addressed their concerns and enhanced the manuscript's readability and solidity.

Reviewer #2 (Remarks to the Author):

The authors have thoroughly addressed my comments and have significantly improved the manuscript through their revisions. I have no further suggestions and recommend the manuscript for publication.

We thank the reviewer for their supportive comments and for recommending our manuscript for publication. We are pleased that the revisions have successfully addressed all points raised.

Reviewer #3 (Remarks to the Author):

We thank the reviewer for their time and contribution to the review of our manuscript.